# Off-Policy Evaluation and Learning
# for External Validity under a Covariate Shift

**Masatoshi Uehara[1]**,*    **Masahiro Kato[2]**,*    **Shota Yasui[2]**
[1] Cornell University
mu223@cornell.edu
[2]CyberAgent Inc.
masahiro_kato@cyberagent.co.jp
yasui_shota@cyberagent.co.jp

## Abstract

We consider evaluating and training a new policy for the evaluation data by using the historical data obtained from a different policy. The goal of *off-policy evaluation* (OPE) is to estimate the expected reward of a new policy over the evaluation data, and that of *off-policy learning* (OPL) is to find a new policy that maximizes the expected reward over the evaluation data. Although the standard OPE and OPL methods assume the same distribution of covariate between the historical and evaluation data, a covariate shift often exists in real-world applications, i.e., the distribution of the covariate of the historical data is different from that of the evaluation data. In this paper, we derive the efficiency bound of an OPE estimator under a covariate shift. Then, we propose doubly robust and efficient estimators for OPE and OPL under a covariate shift by using a nonparametric estimator of the density ratio between the historical and evaluation data distributions. We also discuss other possible estimators and compare their theoretical properties. Finally, we conduct experiments to confirm the effectiveness of the proposed estimators.

## 1 Introduction

In various applications, such as the design of advertisement, personalized medicine, search engines, and recommendation systems, there is a significant interest in evaluating and learning a new policy from historical data (Beygelzimer & Langford, 2009; Li et al., 2010; Athey & Wager, 2017). To accomplish this, we use *off-policy evaluation* (OPE) and *off-policy learning* (OPL) methods. The goal of OPE is to evaluate a new policy by estimating the expected reward of the new policy (Dudík et al., 2011; Wang et al., 2017; Narita et al., 2019; Bibaut et al., 2019; Kallus & Uehara, 2019; Oberst & Sontag, 2019). In contrast, OPL aims to find a new policy that maximizes the expected reward (Zhao et al., 2012; Kitagawa & Tetenov, 2018; Zhou et al., 2018; Chernozhukov et al., 2019).

Although the OPE method provides an estimator of the expected reward of a new policy, most existing studies presume that the distributions of covariates are the same between the historical and evaluation data. However, in many real-world applications, the expected reward of a new policy over the distribution of evaluation data is of significant interest, which can be different from that of historical data. For example, in the medical field, it is known that the results of a randomized controlled trial (RCT) cannot be directly transported because the covariate distribution in a target population is different (Cole & Stuart, 2010). This problem is known as a lack of *external validity* (Pearl & Bareinboim, 2014). These situations, in which historical and evaluation data follow different distributions, are also known as *covariate shifts* (Shimodaira, 2000; Sugiyama et al., 2008). This situation is illustrated in Figure 1.

---

Under a covariate shift, the standard OPE methods do not yield a consistent estimator of the expected reward over the evaluation data. Moreover, a covariate shift changes the efficiency bound of an OPE estimator, which is the lower bound of the asymptotic mean squared error (MSE) among reasonable $\sqrt{n}$-consistent estimators. Besides, standard theoretical analysis of OPE cannot be applied to covariate shift cases as in Remark 2. To handle the covariate shift, we apply importance weighting using the density ratio between the distributions of the covariates of the historical and evaluation data (Shimodaira, 2000; Reddi et al., 2015).

**Contributions:** This paper has four main contributions. First, we derive an efficiency bound of OPE under the covariate shift (Section 3). Second, in Section 4, we propose estimators constructed by the estimators of the density ratio, behavior policy, and conditional expected reward. In particular, we employ nonparametric density ratio estimation (Kanamori et al., 2012) to estimate the density ratio. The proposed estimator is an efficient estimator, which achieves the efficiency bound under mild nonparametric rate conditions of the estimators of nuisance functions. In addition, this estimator is robust to model-misspecification of estimators in the sense that the resulting estimator is consistent if either (i) models of the density ratio and the behavior policy or (ii) a model of the conditional average treatment effect is correct. Importantly, we do not require the Donsker conditions for those estimators by applying the cross-fitting (Section 4). Third, we propose other possible estimators for our problem setting and compare them (Section 5). Fourth, an OPL method is proposed based on the efficient estimators (Section 6). All proofs are shown in Appendix E.

**Related work:** The difference between distributions of covariates conditioned on a chosen action is known as a covariate shift (Zhang et al., 2013b; Johansson et al., 2016). In this paper, a covariate shift refers to the different distributions of covariates between historical and evaluation data. Dahabreh et al. (2019), Johansson et al. (2018), and Sondhi et al. (2020) analyzed the treatment effect estimation under a covariate shift; however, our perspective and analysis are completely different from theirs. Besides, there are many studies regarding the external validity on a causal directed acyclic graph (Pearl & Bareinboim, 2011, 2014). This paper focuses on statistical inference and learning instead of an identification strategy.

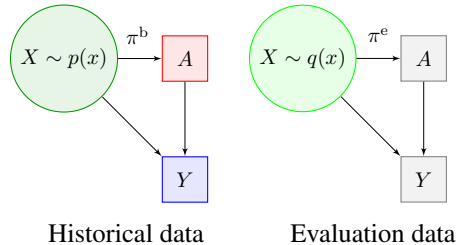

Figure 1: OPE under a covariate shift. The covariate, action, and reward are denoted as $X$, $A$, and $Y$, respectively. The evaluation and behavior policies are denoted as $\pi^{\mathrm{e}}, \pi^{\mathrm{b}}$ respectively. Here, $p(x) \neq q(x)$, and the density ratio $q(x)/p(x)$ is unknown. The density $p(y \mid a, x)$ is the same in historical and evaluation data. For the evaluation data, $A$ and $Y$ are not observed.

## 2 Problem Formulation

In this section, we introduce our problem setting and review the relevant literature.

### 2.1 Data-Generating Process with Evaluation Data

For an individual $i \in \mathbb{N}$, let $A_i$ be an action taking variable in $\mathcal{A}$ and $Y_i \in \mathbb{R}$ be a reward. Let $X_i$ and $Z_i$ be the *covariate* observed by the decision maker when choosing an action, and $\mathcal{X}$ be the space of the covariate. Let a policy $\pi : \mathcal{X} \times \mathcal{A} \to [0, 1]$ be a function of a covariate $x$ and action $a$, which can be considered as the probability of choosing an action $a$ given $x$. In this paper, we have access to *historical* and *evaluation data*. For the historical data, we can observe a dataset $\mathcal{D}^{\mathrm{hst}} = \{(X_i, A_i, Y_i)\}_{i=1}^{n^{\mathrm{hst}}}$, which are *independent and identically distributed* (i.i.d.) for the evaluation data, we can observe an i.i.d. dataset $\mathcal{D}^{\mathrm{evl}} = \{Z_i\}_{i=1}^{n^{\mathrm{evl}}}$, where $n^{\mathrm{hst}}$ and $n^{\mathrm{evl}}$ denote the sample sizes of historical and evaluation data, respectively. We assume $\mathcal{D}^{\mathrm{hst}}$ and $\mathcal{D}^{\mathrm{evl}}$ are independent. Then, we assume the data-generating process (DGP) as $\mathcal{D}^{\mathrm{hst}} = \{(X_i, A_i, Y_i)\}_{i=1}^{n^{\mathrm{hst}}} \sim p(x)\pi^{\mathrm{b}}(a \mid x)p(y \mid x, a)$ and $\mathcal{D}^{\mathrm{evl}} = \{Z_i\}_{i=1}^{n^{\mathrm{evl}}} \sim q(z)$, where $n^{\mathrm{hst}} = \rho n$, $n^{\mathrm{evl}} = (1 - \rho)n$, $p(x)$ and $q(x)$ are densities[2]

over $\mathcal{X}$, and $\rho \in (0,1)$ is a constant. The policy $\pi^{\mathrm{b}}(a \mid x)$ of the historical data is called a *behavior policy*. We generally assume $p(x)$, $q(x)$ and $\pi^{\mathrm{b}}(a \mid x)$ to be unknown. In comparison to the usual OPE, the density of historical data, $p(x)$, can differ from that of the evaluation data, $q(x)$.

**Notation:** This paper distinguishes the covariates between the historical and evaluation data as $X_i$ and $Z_i$, respectively. In addition, for a function $\mu : \mathcal{X} \to \mathbb{R}$, $\mathbb{E}[\mu(X)]$ and $\mathbb{E}[\mu(Z)]$ imply taking expectation over historical and evaluation data, respectively. Likewise, the empirical approximation is denoted as $\mathbb{E}_{n^{\mathrm{hst}}}[\mu(X)] = 1/n^{\mathrm{hst}} \sum_i \mu(X_i)$ and $\mathbb{E}_{n^{\mathrm{evl}}}[\mu(X)] = 1/n^{\mathrm{evl}} \sum_i \mu(Z_i)$. Additionally, let $\|\mu(X, A, Y)\|_2$ be $\mathbb{E}[\mu^2(X, A, Y)]^{1/2}$ for the function $\mu$, $\mathbb{E}_{p(x,a,y)}[\mu(x,a,y)]$ be $\int \mu(x,a,y)p(x,a,y)\mathrm{d}(x,a,y)$, the asymptotic MSE of estimator $\hat{R}$ be $\mathrm{Asmse}[\hat{R}] = \lim_{n\to\infty} n\mathbb{E}[(\hat{R} - R)^2]$, and $\mathcal{N}(0, A)$ be a normal distribution with mean 0 and variance $A$. In addition, we use functions $r(x) = q(x)/p(x)$, $w(a,x) = \pi^{\mathrm{e}}(a \mid x)/\pi^{\mathrm{b}}(a \mid x)$, and $f(a,x) = \mathbb{E}[Y \mid X = x, A = a]$. Let us denote the estimators of $r(x)$, $w(a,x)$, and $f(a,x)$ as $\hat{r}(x)$, $\hat{w}(a,x)$, and $\hat{f}(a,x)$, respectively. Other notations are summarized in Appendix A.

**Remark 1.** Although we do not explicitly use counter-factual notation (Rubin, 1987), if we assume the usual conditions, our results immediately apply (Appendix B).

## 2.2 Off-Policy Evaluation and Learning

We are interested in estimating the expected reward of an *evaluation policy* $\pi^{\mathrm{e}}(a \mid x)$, which is prespecified for the evaluation data. Here, we assume a *covariate shift*, which is a common situation in the literature of external validity. Under a covariate shift, the conditional distribution of $y$ is the same between the historical and evaluation data, whereas the distribution of evaluation data is different from that of historical data, i.e., the distribution of evaluation data with evaluation policy $\pi^{\mathrm{e}}$ follows $q(z)\pi^{\mathrm{e}}(a \mid z)p(y \mid a, z)$. Then, we define the expected reward of the evaluation policy as

$$R(\pi^{\mathrm{e}}) := \mathbb{E}_{q(z)\pi^{\mathrm{e}}(a|z)p(y|a,z)} [y] . \tag{1}$$

The first goal is OPE; i.e., estimating $R(\pi^{\mathrm{e}})$ using the historical data $\{X_i, A_i, Y_i\}_{i=1}^{n^{\mathrm{hst}}}$ and evaluation data $\{Z_i\}_{i=1}^{n^{\mathrm{evl}}}$. The second goal is OPL; i.e., training a new policy that maximizes the expected reward as $\pi^* = \arg\max_{\pi \in \Pi} R(\pi)$, where $\Pi$ is the policy class. In some cases, to construct an estimator $R(\pi)$, we use $r(x)$, $w(a,x)$, and $f(a,x)$. These functions are known as *nuisance functions*. Let $\hat{r}(x)$, $\hat{w}(a,x)$, and $\hat{f}(a,x)$ be their estimators.

**Assumptions:** We assume strong overlaps for $r(x)$, $w(a,x)$ and theirs estimators and boundedness for $Y_i$ and $\hat{f}$ using a constant $R_{\max} > 0$.

**Assumption 1.** $0 \leq r(x) \leq C_1$, $0 \leq w(a,x) \leq C_2$, $0 \leq Y_i \leq R_{\max}$.

**Assumption 2.** $0 \leq \hat{r}(x) \leq C_1$, $0 \leq \hat{w}(a,x) \leq C_2$, $0 \leq \hat{f}(a,x) \leq R_{\max}$.

## 2.3 Preliminaries

Here, we review existing work of OPE, OPL, and the density ratio estimation.

**Standard OPE and OPL:** We review three types of standard estimators of $\mathbb{E}_{p(x)\pi^{\mathrm{e}}(a|x)p(y|x,a)}[y]$ for the case where $q(x) = p(x)$ in (1). The first estimator is an inverse probability weighting (IPW) estimator given by $\mathbb{E}_{n^{\mathrm{hst}}}[\hat{w}(A, X)Y]$ (Horvitz & Thompson, 1952; Rubin, 1987; Cheng, 1994; Hirano et al., 2003b; Swaminathan & Joachims, 2015b). Even though this estimator is unbiased when the behavior policy is known, it often suffers from high variance. The second estimator is a direct method (DM) estimator $\mathbb{E}_{n^{\mathrm{hst}}}[\hat{f}(A, X)]$ (Hahn, 1998), which is weak against model misspecification for $f(a,x)$. The third estimator is a doubly robust estimator (Robins et al., 1994) defined as

$$\mathbb{E}_{n^{\mathrm{hst}}}[\hat{w}(A, X)\{Y - \hat{f}(A, X)\} + \mathbb{E}_{\pi^{\mathrm{e}}(a|X)}[\hat{f}(a, X) \mid X]]. \tag{2}$$

Under certain conditions, this estimator is known to achieve the efficiency bound (a.k.a semiparametric lower bound), which is the lower bound of the asymptotic MSE of OPE, among regular

$\sqrt{n}$-consistent estimators (van der Vaart, 1998, Theorem 25.20) [3]. This efficiency bound is

$$\mathbb{E}[w^2(A, X)\mathrm{var}[Y \mid A, X]] + \mathrm{var}[v(X)], \tag{3}$$

where $v(x) = \mathbb{E}_{\pi^{\mathrm{e}}(a|x)}[f(a, x) \mid x]$ (Narita et al., 2019). Such an estimator is called an *efficient estimator*. These estimators are also used for OPL (Zhang et al., 2013a; Athey & Wager, 2017).

**Remark 2** (Difference from standard OPE problems)**.** Our current problem, i.e., policy evaluation *under a shift in domain and policy*, differs from a standard policy evaluation problem *only under a shift in the policy*. For our domain and policy shift problem, we assume a *stratified sampling*, i.e, fixed $\rho$ w.r.t $n$. Instead, in the literature of a policy shift, people assume a sampling scheme is i.i.d. As indicated by Wooldridge (2001), the difference of these two sampling schemes results in the analysis being different.

With respect to our problem, we can also assume that samples are i.i.d. by considering $\rho$ to be a random variable and by assuming each replication follows a *mixture distribution* (Dahabreh et al., 2019). However, under this assumption, the efficiency bound cannot be calculated explicitly. In addition, $\rho$ is often defined as a constant value by some design (Qin, 1998).

**Density Ratio Estimation:** To estimate $R(\pi^{\mathrm{e}})$, we apply importance weighting using the density ratio between the distributions of historical and evaluation covariates. For example, if we know $r(x)$ and $w(a, x)$, we can construct an estimator of $R(\pi^{\mathrm{e}})$ as $\mathbb{E}_{n^{\mathrm{hst}}}[r(X)w(A, X)Y]$. If we know the behavior policy as in an RCT, we can exactly know $w(a, x)$. However, because we do not know the density ratio $r(x)$ directly even in an RCT, we have to estimate $r(x)$ using the covariate data $\{X_i\}_{i=1}^{n^{\mathrm{hst}}}$ and $\{Z_i\}_{i=1}^{n^{\mathrm{evl}}}$. To estimate the density ratio $r(x)$, we use a nonparametric one-step loss based estimator. For example, we employ *Least-Squares Importance Fitting* (LSIF), which uses the squared loss to fit the density-ratio function (Kanamori et al., 2012). We show details in Appendix C.

## 3   Efficiency Bound under a Covariate Shift

We discuss the efficiency bound of OPE under a covariate shift. An efficiency bound is defined for an estimand under some posited models of the DGP (Bickel et al., 1998). If this posited model is a parametric model, it is equal to the Cramér-Rao lower bound. When this posited model is non or semiparametric model, we can still define a corresponding Cramér-Rao lower bound. In this paper, we modify the standard theory under i.i.d. sampling to the current problem assuming a stratified sampling scheme. The formal definition is shown in Appendix D.

Here, we show the efficiency bound of OPE under a covariate shift.

**Theorem 1.** *The efficiency bound of $R(\pi^{\mathrm{e}})$ under fully nonparametric models is*

$$\Upsilon(\pi^{\mathrm{e}}) = \rho^{-1}\mathbb{E}[r^2(X)w^2(A, X)\mathrm{var}[Y \mid A, X]] + (1 - \rho)^{-1}\mathrm{var}[v(Z)], \tag{4}$$

*where $v(z) = \mathbb{E}_{\pi^{\mathrm{e}}(a|z)}[f(a, z) \mid z]$. The efficiency bound under a nonparametric model with fixed $p(x)$ and $\pi^{\mathrm{b}}(a \mid x)$ is the same.*

Three things are remarked. First, knowledge of the densities of the historical data $p(x)$ and the behavior policy $\pi^{\mathrm{b}}(a \mid x)$ does not change the efficiency bound (3). This is because the target functional does not include these two densities. Second, the efficiency bound under a covariate shift (4) reduces to the bound without a covariate shift (3) in a special case, $r(x) = 1$ and $\rho = 0.5$. Then, we can see (4)= 2×(3). The factor 2 originates from the scaling of the asymptotic MSE. Third, we need to calculate the *efficient influence function*, which is a key function for deriving the efficiency bound. This function is useful for constructing an efficient estimator.

## 4   OPE under a Covariate Shift

For OPE under a covariate shift, we propose an estimator constructed from the following basic form:

$$\mathbb{E}_{n^{\mathrm{hst}}}[\hat{r}(X)\hat{w}(A, X)\{Y - \hat{f}(A, X)\}] + \mathbb{E}_{n^{\mathrm{evl}}}[\hat{v}(Z)], \tag{5}$$

**Algorithm 1** Doubly Robust Estimator under a Covariate Shift

---

**Input**: The evaluation policy $\pi^{\mathrm{e}}$.

Take a $\xi$-fold random partition $(I_k)_{k=1}^{\xi}$ of observation indices $[n^{\mathrm{hst}}] = \{1, \ldots, n^{\mathrm{hst}}\}$ such that the size of each fold $I_k$ is $n_k^{\mathrm{hst}} = n^{\mathrm{hst}}/\xi$.

Take a $\xi$-fold random partition $(J_k)_{k=1}^{\xi}$ of observation indices $[n^{\mathrm{evl}}] = \{1, \ldots, n^{\mathrm{evl}}\}$ such that the size of each fold $J_k$ is $n_k^{\mathrm{evl}} = n^{\mathrm{evl}}/\xi$.

For each $k \in [\xi] = \{1, \ldots, \xi\}$, define $I_k^c := \{1, \ldots, n^{\mathrm{hst}}\} \setminus I_k$ and $J_k^c := \{1, \ldots, n^{\mathrm{evl}}\} \setminus J_k$.

Define $(\mathcal{S}_k)_{k=1}^{\xi}$ with $\mathcal{S}_k = \{\{(X_i, A_i, Y_i)\}_{i \in I_k^c}, \{Z_j\}_{j \in J_k^c}\}$.

**for** $k \in [\xi]$ **do**

    Construct estimators $\hat{w}_k(a, x)$, $\hat{r}_k(x)$, and $\hat{f}_k(a, x)$ using $\mathcal{S}_k$.

    Construct an estimator $\hat{R}_k$ defined as (6).

**end for**

Construct an estimator $\hat{R}$ of $R$ by taking the average of $\hat{R}_k$ for $k \in [\xi]$, i.e., $\hat{R} = \frac{1}{\xi} \sum_{k=1}^{\xi} \hat{R}_k$.

---

where $\hat{r}(x)$, $\hat{w}(a, x)$, and $\hat{f}(a, x)$ are nuisance estimators of $r(x)$, $w(a, x)$, and $f(a, x)$, and $\hat{v}(z) = \mathbb{E}_{\pi^{\mathrm{e}}(a|z)}[\hat{f}(a, z) \mid z]$. As well as the standard doubly robust estimator (2), the above form is designed to have the double robust structure regarding the model specifications of $r(x)w(a, x)$ and $f(a, x)$. First, we consider the case where $\hat{r}(x) = r(x)$ and $\hat{w}(a, x) = w(a, x)$, but $\hat{f}(a, x)$ is equal to $f^{\dagger}(a, x)$ and different from $f(a, x)$, i.e., we have correct models for $r(x)$ and $w(a, x)$, but not for $f(a, x)$. Then, (5) is a consistent estimator of $R(\pi^{\mathrm{e}})$ because

$$\mathbb{E}_{n^{\mathrm{hst}}}[r(X)w(A, X)Y] + \mathbb{E}_{n^{\mathrm{evl}}}[\mathbb{E}_{\pi^{\mathrm{e}}(a|Z)}[f^{\dagger}(a, Z) \mid Z]] - \mathbb{E}_{n^{\mathrm{hst}}}[r(X)w(A, X)f^{\dagger}(A, X)]$$
$$\approx \mathbb{E}_{n^{\mathrm{hst}}}[r(X)w(A, X)Y] + 0 \approx R(\pi^{\mathrm{e}}).$$

Second, we consider the case where $\hat{f}(a, x) = f(a, x)$, but $\hat{r}(x)$ and $\hat{w}(a, x)$ are equal to functions $r^{\dagger}(x)$ and $w^{\dagger}(a, x)$, which are different from $r(x)$ and $w(a, x)$, respectively, i.e, we have correct models for $f(a, x)$, but not for $r(x)$ and $w(a, x)$. Then, (5) is a consistent estimator for $R(\pi^{\mathrm{e}})$ because

$$\mathbb{E}_{n^{\mathrm{hst}}}[r^{\dagger}(X)w^{\dagger}(a, x)\{Y - f(A, X)\}] + \mathbb{E}_{n^{\mathrm{evl}}}[\mathbb{E}_{\pi^{\mathrm{e}}(a|Z)}[f(a, Z) \mid Z]]$$
$$\approx \mathbb{E}_{n^{\mathrm{evl}}}[\mathbb{E}_{\pi^{\mathrm{e}}(a|Z)}[f(a, Z) \mid Z]] + 0 \approx R(\pi^{\mathrm{e}}).$$

The formal result is given later in Theorem 3.

Next, we consider estimating $r(x)$, $w(a, x)$, and $f(a, x)$. For example, for $f(a, x)$ and $w(a, x)$, we can apply complex and data-adaptive regression and density estimation methods such as random forests, neural networks, and highly adaptive Lasso (Díaz, 2019). Note that $\hat{w}(a, x)$ is estimated as $\pi^{\mathrm{e}}/\hat{\pi}^b$ because $\pi^{\mathrm{e}}$ is known, where $\hat{\pi}^b$ is an estimator of $\pi^b$. For $r(x)$, we can use the data-adaptive density ratio method in Section 2.3. Although such complex estimators approximate the true values well, it is pointed out that such estimators often violate the Donsker condition (van der Vaart, 1998; Chernozhukov et al., 2018). [4], which is required to obtain the asymptotic distribution of an estimator of interest, such as (5).

To derive the asymptotic distributions of an estimator of $R(\pi^{\mathrm{e}})$ using estimators without the Donsker condition, we apply cross-fitting (Klaassen, 1987; Zheng & van der Laan, 2011; Chernozhukov et al., 2018) based on (5). The procedure is as follows. First, we separate data $\mathcal{D}^{\mathrm{hst}}$ and $\mathcal{D}^{\mathrm{evl}}$ into $\xi$ groups. Next, using samples in each group, we estimate the nuisance functions nonparametrically. Then, we construct an estimator of $R(\pi^{\mathrm{e}})$ using the nuisance estimators. For each group $k \in \{1, 2, \ldots, \xi\}$, we define

$$\hat{R}_k = \mathbb{E}_{n_k^{\mathrm{hst}}}[\hat{r}^{(k)}(X)\hat{w}^{(k)}(A, X)\{Y - \hat{f}^{(k)}(A, X)\}] + \mathbb{E}_{n_k^{\mathrm{evl}}}[\mathbb{E}_{\pi^{\mathrm{e}}}[\hat{f}^{(k)}(a, Z)|Z]], \quad (6)$$

where $\mathbb{E}_{n_k^{\mathrm{hst}}}$ is the sample average over the $k$-th partitioned historical data with $n_k^{\mathrm{hst}}$ samples and $\mathbb{E}_{n_k^{\mathrm{evl}}}$ is the sample average over the $k$-th partitioned evaluation data with $n_k^{\mathrm{evl}}$ samples. Finally, we

construct an estimator of $R(\pi^{\mathrm{e}})$ by taking the average of the the $K$ estimators, $\{\hat{R}_k\}_{k=1}^{K}$. We call the estimator *doubly robust estimator under a covariate shift* (DRCS) and denote it as $\hat{R}_{\mathrm{DRCS}}(\pi^{\mathrm{e}})$. The entire procedure is given in Algorithm 1.

In the following, we show the asymptotic property of $\hat{R}_{\mathrm{DRCS}}(\pi^{\mathrm{e}})$. First, $\hat{R}_{\mathrm{DRCS}}(\pi^{\mathrm{e}})$ is efficient.

**Theorem 2** (Efficiency). *For* $k \in \{1, \cdots, \xi\}$, *assume* $\alpha\beta = \mathrm{o}_p(n^{-1/2}), \alpha = \mathrm{o}_p(1), \beta = \mathrm{o}_p(1)$ *where* $\|\hat{r}^{(k)}(X)\hat{w}^{(k)}(A, X) - r(X)w(A, X)\|_2 = \alpha, \|\hat{f}^{(k)}(A, X) - f(A, X)\|_2 = \beta$. *Then,* $\sqrt{n}(\hat{R}_{\mathrm{DRCS}}(\pi^{\mathrm{e}}) - R(\pi^{\mathrm{e}})) \xrightarrow{d} \mathcal{N}(0, \Upsilon(\pi^{\mathrm{e}}))$, *where* $\Upsilon(\pi^{\mathrm{e}})$ *is the efficiency bound in Theorem 1.*

Importantly, the Donsker condition is *not* needed for nuisance estimators owing to the cross-fitting and the doubly robust form of $\hat{R}_{\mathrm{DRCS}}$. In this regard, our only requirement is the rate conditions, which are mild because these are nonparametric rates smaller than $1/2$. For example, this is satisfied when $\alpha = \beta = \mathrm{o}_p(n^{-1/4})$. With some smoothness conditions, the nonparametric estimator $\hat{f}(a, x)$ can achieve this convergence rate (Wainwright, 2019). Regarding $r(x)w(a, x)$, we can show that if $\hat{r}(x)$ and $\hat{w}(a, x)$ similarly satisfy certain nonparametirc rates, $\hat{r}(x)\hat{w}(a, x)$ satisfies it as well.

**Lemma 1.** *Assume* $\|\hat{r}(X) - r(X)\|_2 = \mathrm{o}_p(n^{-p})$ *and* $\|\hat{w}(A, X) - w(A, X)\|_2 = \mathrm{o}_p(n^{-p})$. *Then,* $\|\hat{r}(X)\hat{w}(A, X) - r(X)w(A, X)\|_2 = \mathrm{o}_p(n^{-p})$.

Next, we formally show the double robustness of the estimator, i.e., the estimator is consistent if either $r(x)w(a, x)$ or $f(a, x)$ is correct.

**Theorem 3** (Double robustness). *For* $k \in \{1, \cdots, \xi\}$, *assume that* $\exists f^{\dagger}, r^{\dagger}, w^{\dagger}, \|\hat{f}^{(k)}(A, X) - f^{\dagger}(A, X)\|_2 = \mathrm{o}_p(1)$ *and* $\|\hat{r}^{(k)}(X)\hat{w}^{(k)}(A, X) - r^{\dagger}(X)w^{\dagger}(A, X)\|_2 = \mathrm{o}_p(1)$. *If* $r^{\dagger}(x)w^{\dagger}(a, x) = r(x)w(a, x)$ *or* $q^{\dagger}(a, x) = q(a, x)$ *holds, the estimator* $\hat{R}_{\mathrm{DRCS}}(\pi^{\mathrm{e}})$ *is consistent.*

In a standard OPE, the DR type estimator is consistent when we know the behavior policy. In contrast, under a covariate shift, even when the behavior policy is known, we cannot claim that $\hat{R}_{\mathrm{DRCS}}(\pi^{\mathrm{e}})$ is consistent because $r(x)$ is unknown. This result suggests the estimation of $r(x)$ is crucial.

**Remark 3** (OPE with Known Distribution of Evaluation Data). As a special case of OPE under a covariate shift, we consider a case where $q(x)$ is known. This case can be regarded as a standard OPE situation by regarding $p(x)\pi^{\mathrm{e}}(a \mid x)$ as the behavior policy, the evaluation policy as $q(x)\pi^{\mathrm{e}}(a \mid x)$, and $(A, X)$ as the action. The details of this setting is shown in Appendix F

**Remark 4** (Relation with Pearl & Bareinboim (2014)). A transport formula (Pearl & Bareinboim, 2014, (3.1)) essentially leads to the DM estimator $\mathbb{E}_{n^{\mathrm{evl}}}[\hat{v}(Z)]$. Though they propose a general identification strategy, they do not discuss how to conduct efficient estimation given finite samples.

**Remark 5** (Construction of $\hat{R}_{\mathrm{DRCS}}(\pi^{\mathrm{e}})$). We construct $\hat{R}_{\mathrm{DRCS}}(\pi^{\mathrm{e}})$ so that it has a doubly robust structure. The construction is also motivated by the efficient influence function. More specifically, this estimator is introduced by plugging the nuisance estimators into the efficient influence function.

## 5  Other Candidates of Estimators

We have discussed the doubly robust estimator in the previous section. Next, we propose other estimators under a covariate shift based on the IPW and DM estimators. We analyze the property of each estimator with the nuisance estimators obtained from the classical kernel regression (Nadaraya, 1964; Watson, 1964). We show regularity conditions and formal results of Theorems 4–6 in Appendix E.

### 5.1  IPW Estimators and DM Estimator

We consider IPW and DM type estimators under a covariate shift for *each case* where we have an oracle of $\pi^{\mathrm{b}}(a \mid x)$ and we do not have any oracles of nuisance functions, *respectively*. In comparison to a standard OPE case, we can consider two fundamentally different IPW type estimators.

**IPW estimator with oracle** $\pi^{\mathrm{b}}(x)$**:** This is a natural setting in an RCT and and A/B testing because we assign actions following a certain probability in theses cases. Let us define an IPW estimator under a covariate shift with the true behavior policy $\pi^{\mathrm{b}}(a \mid x)$ (IPWCSB) as $\hat{R}_{\mathrm{IPWCSB}}(\pi^{\mathrm{e}}) =$

Table 1: Comparison of estimators. The parentheses means that efficiency is ensured when using specific estimators for nuisances, such as kernel estimators. Non-Donsker means whether any non-Donsker type complex estimators can be allowed to plug-in with a valid theoretical guarantee. All of the estimators here do not require any parametric model assumptions.

| Estimator | Efficiency | Double Robustness | Nuisance Functions | Without Oracle of $\pi^{\mathrm{b}}(x)$ | Non-Donsker |
|---|---|---|---|---|---|
| $\hat{R}_{\mathrm{IPWCSB}}(\pi^{\mathrm{e}})$ | | | $r$ | | |
| $\hat{R}_{\mathrm{IPWCS}}(\pi^{\mathrm{e}})$ | $(\checkmark)$ | | $r, w$ | $\checkmark$ | |
| $\hat{R}_{\mathrm{DM}}(\pi^{\mathrm{e}})$ | $(\checkmark)$ | | $f$ | $\checkmark$ | |
| $\hat{R}_{\mathrm{DRCS}}(\pi^{\mathrm{e}})$ | $\checkmark$ | $\checkmark$ | $r, w, f$ | $\checkmark$ | $\checkmark$ |

$\mathbb{E}_{n^{\mathrm{hst}}} \left[ \frac{\hat{q}(X)}{\hat{p}(X)} \frac{\pi^{\mathrm{e}}(A|X)Y}{\pi^{\mathrm{b}}(A|X)} \right]$. For example, we use a classical kernel density estimators of $q(x)$ and $p(x)$ defined as $\hat{q}_h(x) = \frac{1}{n^{\mathrm{evl}}} \sum_{i=1}^{n^{\mathrm{evl}}} h^{-d} K\left(\frac{Z_i - x}{h^d}\right)$ and $\hat{p}_h(x) = \frac{1}{n^{\mathrm{hst}}} \sum_{i=1}^{n^{\mathrm{hst}}} h^{-d} K\left(\frac{X_i - x}{h^d}\right)$, where $K(\cdot)$ is a kernel function, $h$ is the bandwidth of $K(\cdot)$, and $d$ is a dimension of $x$. When using a kernel estimator, we obtain the following theorem.

**Theorem 4** (Informal). *When $\hat{q}(x) = \hat{q}_h(x)$, $\hat{p}(x) = \hat{p}_h(x)$, the asymptotic MSE of $\hat{R}_{\mathrm{IPWCSB}}(\pi^{\mathrm{e}})$ is $\rho^{-1}\mathrm{var}[r(X)\{w(A, X)Y - v(X)\}] + (1 - \rho)^{-1}\mathrm{var}[v(Z)]$.*

**Fully nonparametric IPW estimator:**   Next, *for the case without the oracle* $\pi^{\mathrm{b}}$, let us define an IPW estimator under a covariate shift (IPWCS) as $\hat{R}_{\mathrm{IPWCS}}(\pi^{\mathrm{e}}) = \mathbb{E}_{n^{\mathrm{hst}}} \left[ \frac{\hat{q}(X)\pi^{\mathrm{e}}(A|X)Y}{\hat{p}(X)\hat{\pi}^{\mathrm{b}}(A|X)} \right]$. This estimator achieves the efficiency bound.

**Theorem 5** (Informal). *When $\hat{q}(x) = \hat{q}_h(x)$, $\hat{p}(x) = \hat{p}_h(x)$ and $\hat{\pi}^{b}(a \mid x) = \hat{\pi}_h^{b}(a \mid x)$, where $\hat{\pi}_h^{b}(a \mid x)$ is a kernel estimator based on $\mathcal{D}^{\mathrm{hst}}$, the asymptotic MSE of $\hat{R}_{\mathrm{IPWCS}}(\pi^{\mathrm{e}})$ is $\Upsilon(\pi^{\mathrm{e}})$.*

**DM Estimator:**   Finally, we define a nonparametric DM estimator $\hat{R}_{\mathrm{DM}}(\pi^{\mathrm{e}})$ as $\mathbb{E}_{n^{\mathrm{evl}}}[\mathbb{E}_{\pi^{\mathrm{e}}(a|Z)}[\hat{f}(a, Z) \mid Z]]$. This estimator achieves the efficiency bound.

**Theorem 6** (Informal). *When $\hat{f}_h(a, x)$ is a kernel estimator based on $\mathcal{D}^{\mathrm{hst}}$, the asymptotic MSE of $\hat{R}_{\mathrm{DM}}(\pi^{\mathrm{e}})$ is $\Upsilon(\pi^{\mathrm{e}})$.*

## 5.2   Comparison of Estimators

We compare the estimators discussed so far. This discussion is summarized in Table 1. First, the estimator $\hat{R}_{\mathrm{DRCS}}$ allows any non-Donsker type complex estimators with lax convergence rate conditions of the nuisance estimators. However, the analyses of $\hat{R}_{\mathrm{IPWCS}}$ and $\hat{R}_{\mathrm{DM}}$ are specific to the kernel estimators though the asymptotic MSE of $\hat{R}_{\mathrm{IPWCS}}$, $\hat{R}_{\mathrm{DM}}$, and $\hat{R}_{\mathrm{DRCS}}$ are the same in this special case. When the kernel estimators are replaced with any non-Donsker type complex estimators, the rate condition $\|\hat{r}(X)\hat{w}(A, X) - r(X)w(A, X)\|_2 = \mathrm{o}_p(n^{-1/4})$ or $\|\hat{f}(A, X) - f(A, X)\|_2 = \mathrm{o}_p(n^{-1/4})$ *cannot* guarantee the $\sqrt{n}$-consistency and efficiency even if we use cross-fitting. Therefore, we cannot show asymptotic normality for IPW and DM type estimators, even if applying cross-fitting. The fact that the bias of DR type estimator is reduced to the product term of two convergence rates has a critical role. Second, the only $\hat{R}_{\mathrm{DRCS}}$ has double robustness; however, $\hat{R}_{\mathrm{IPWCS}}$ and $\hat{R}_{\mathrm{DM}}$ do not have this property.

**Comparison among IPW estimators:**   We observe that the asymptotic MSE of $\hat{R}_{\mathrm{IPWCS}}$[5] is smaller than that of $\hat{R}_{\mathrm{IPWCSB}}$. This result looks unusual because $\hat{R}_{\mathrm{IPWCSB}}$ uses more knowledge than $\hat{R}_{\mathrm{IPWCS}}$. The intuitive reason for this fact is that $\hat{R}_{\mathrm{IPWCS}}$ is considered to be using control variate. The same paradox is known in other works of causal inference (Robins et al., 1992). Note that this fact does not imply $\hat{R}_{\mathrm{IPWCS}}$ is superior to $\hat{R}_{\mathrm{IPWCSB}}$ because smoothness conditions are required in $\hat{R}_{\mathrm{IPWCS}}$, and this can be violated in practice (Robins & Ritov, 1997).

# 6 OPL under a Covariate Shift

In this section, we propose an OPL method based on the doubly robust estimator $\hat{R}_{\mathrm{DRCS}}(\pi^{\mathrm{e}})$ to estimate the optimal policy that maximizes the expected reward over the evaluation data. Note that the optimal policy $\pi^*$ is defined as $\pi^* = \arg\max_{\pi \in \Pi} R(\pi)$. By applying each OPE estimator, we can define the following estimators: $\hat{\pi}_{\mathrm{DRCS}} = \arg\max_{\pi \in \Pi} \hat{R}_{\mathrm{DRCS}}(\pi)$, $\hat{\pi}_{\mathrm{DM}} = \arg\max_{\pi \in \Pi} \hat{R}_{\mathrm{DM}}(\pi)$, and $\hat{\pi}_{\mathrm{IPWCS}} = \arg\max_{\pi \in \Pi} \hat{R}_{\mathrm{IPWCS}}(\pi)$. To obtain a theoretical implication, for simplicity, we assume $\mathcal{A}$ is a finite state space, and the policy class $\Pi$ is fixed. Then, for the $\epsilon$-Hamming covering number $N_H(\epsilon, \Pi)$ and its entropy integral $\kappa(\Pi) := \int_0^\infty \sqrt{\log N_H(\epsilon^2, \Pi)} \mathrm{d}\epsilon$ (Zhou et al., 2018), the regret bound of $\hat{\pi}_{\mathrm{DRCS}}$ is obtained.

**Theorem 7** (Regret bound of $\hat{\pi}_{\mathrm{DRCS}}$). *Assume that for any $0 < \epsilon < 1$, there exists $\omega$ such that $N_H(\epsilon, \Pi) = \mathcal{O}(\exp(1/\epsilon)^\omega), 0 < \omega < 0.5$. Also suppose that for $k \in \{1, \cdots, \xi\}$, $\|\hat{r}^{(k)}(X) - r(X)\|_2 = \mathrm{o}_p(n^{-1/4})$, $\|1/\hat{\pi}^{(k)\mathrm{b}}(A, X) - 1/\pi^{\mathrm{b}}(A, X)\|_2 = \mathrm{o}_p(n^{-1/4})$, and $\|\hat{f}^{(k)}(A, X) - f(A, X)\|_2 = \mathrm{o}_p(n^{-1/4})$. Then, by defining $\Upsilon_* = \sup_{\pi \in \Pi} \Upsilon(\pi)$, there exists an integer $N_\delta$ such that with probability at least $1 - 2\delta$, for all $n \geq N_\delta$,*

$$R(\pi^*) - R(\hat{\pi}_{\mathrm{DRCS}}) = \mathcal{O}((\kappa(\Pi) + \sqrt{\log(1/\delta)})\sqrt{\tfrac{\Upsilon_*}{n}}).$$

In comparison to the standard regret results in Swaminathan & Joachims (2015b) and Kitagawa & Tetenov (2018), we do not assume we know the true behavior policy. Because $\hat{R}_{\mathrm{DRCS}}(\pi)$ has a double robust structure, we can obtain the regret bound under weak nonparametric rate conditions without assuming the behavior policy is known. Besides, this theorem shows that the variance term is related to attain the low regret. This is achieved by using the efficient estimator $\hat{R}_{\mathrm{DRCS}}(\pi)$.

# 7 Experiments

In this section, we demonstrate the effectiveness of the proposed estimators using data obtained with bandit feedback. Following previous work (Dudík et al., 2011; Farajtabar et al., 2018), we evaluate the proposed estimators using the standard classification datasets from the UCI repository by transforming the classification data into contextual bandit data. From the UCI repository, we use the `satimage`, `vehicle`, and `pendigits` datasets [6]. The results of the `pendigits` dataset is shown in Appendix H. For each dataset, we randomly choose 800 samples (the results with other sample sizes are reported in Appendix H). First, we classify data into historical and evaluation data with probability defined as $p(hist = +1|X_i) = \frac{C_{\mathrm{prob}}}{1 + \exp(-\tau(X_i) + 0.1\varepsilon)}$, where $hist = +1$ denotes that the sample $i$ belongs to the historical data, $C_{\mathrm{prob}}$ is a constant, $\varepsilon$ is a random variable that follows the standard normal distribution, $X_{k,i}$ is the $k$-th element of the vector $X_i$, and $\tau(X_i) = \sum_{j=1}^5 X_{j,i}$. By adjusting $C_{\mathrm{prob}}$, we classify 70% samples as the historical data and 30% samples as the evaluation data. Thus, we generate historical and evaluation data under a covariate shift. Then, we make a deterministic policy $\pi_d$ by training a logistic regression classifier on the historical data. We construct three different behavior policies as mixtures of $\pi^d$ and the uniform random policy $\pi^u$ by changing a mixture parameter $\alpha$, i.e., $\pi^{\mathrm{b}} = \alpha\pi^d + (1-\alpha)\pi^u$. The candidates of the mixture parameter $\alpha$ are $\{0.7, 0.4, 0.0\}$ as Kallus & Uehara (2019). In Section 7.1, we show the experimental results of OPE. In Section 7.2, we show the experimental results of OPL. In both sections, the historical $p(x)$ and evaluation distributions $q(x)$ are unknown, and the behavior policy $\pi^{\mathrm{b}}$ is also unknown. More details, such as the description of the data and choice of hyperparameters, are in Appendix H.

## 7.1 Experiments of Off-Policy Evaluation

For OPE, we use an evaluation policy $\pi^{\mathrm{e}}$ defined as $0.9\pi^d + 0.1\pi^u$. Here, we compare the MSEs of five estimators, DRCS, DM, DM-R, IPWCS, and IPWCS-R. DRCS is the proposed estimator $\hat{R}_{\mathrm{DRCS}}$ where we use kernel Ridge regression for estimating $f(a, x)$ and $w(a, x)$ and use KuLISF (Kanamori et al., 2012) for $r(x)$. For this estimator, we use 2-fold cross-fitting. DM denotes the direct method estimator $\hat{R}_{\mathrm{DM}}(\pi^{\mathrm{e}})$ with $f(a, x)$ estimated by Nadaraya-Watson regression defined in

Table 2: OPE results. Each (a),(b),(c) refers to the cases where the behavior policies are (a) $0.7\pi^d + 0.3\pi^u$, (b) $0.4\pi^d + 0.6\pi^u$, (c) $0.0\pi^d + 1.0\pi^u$, respectively. The notation – means each value is larger than $1.0$.

OPE with the `satimage` dataset

|  | DRCS MSE SD | IPWCS MSE SD | DM MSE SD | IPWCS-R MSE SD | DM-R MSE SD |
|---|---|---|---|---|---|
| (a) | 0.107 0.032 | – – | **0.042** 0.043 | **0.045** 0.049 | 0.073 0.023 |
| (b) | **0.096** 0.025 | – – | 0.134 0.052 | **0.093** 0.069 | 0.177 0.033 |
| (c) | **0.154** 0.051 | – – | 0.336 0.079 | **0.022** 0.026 | 0.372 0.050 |

OPE with the `vehicle` dataset

|  | DRCS MSE SD | IPWCS MSE SD | DM MSE SD | IPWCS-R MSE SD | DM-R MSE SD |
|---|---|---|---|---|---|
| (a) | **0.029** 0.019 | – – | **0.038** 0.035 | 0.568 0.319 | 0.040 0.014 |
| (b) | **0.019** 0.024 | – – | 0.095 0.062 | 0.576 0.357 | **0.089** 0.019 |
| (c) | **0.037** 0.030 | – – | 0.213 0.049 | 0.233 0.193 | **0.210** 0.031 |

Table 3: OPL results. The alphabets (a),(b), and (c) refer to the cases where the behavior policies are (a) $0.7\pi^d + 0.3\pi^u$, (b) $0.4\pi^d + 0.6\pi^u$, (c) $0.0\pi^d + 1.0\pi^u$, respectively.

OPL with the `satimage` dataset

|  | DRCS RWD | SD | IPWCS RWD | SD | DM RWD | SD |
|---|---|---|---|---|---|---|
| (a) | **0.723** | 0.035 | 0.423 | 0.063 | 0.658 | 0.045 |
| (b) | **0.710** | 0.035 | 0.482 | 0.096 | 0.641 | 0.048 |
| (c) | **0.652** | 0.046 | 0.460 | 0.131 | 0.465 | 0.070 |

OPL with the `vehicle` dataset

|  | DRCS RWD | SD | IPWCS RWD | SD | DM RWD | SD |
|---|---|---|---|---|---|---|
| (a) | **0.496** | 0.017 | 0.310 | 0.030 | 0.411 | 0.040 |
| (b) | **0.510** | 0.029 | 0.290 | 0.051 | 0.393 | 0.052 |
| (c) | **0.480** | 0.044 | 0.280 | 0.041 | 0.313 | 0.065 |

Section 5. DM-R is the same estimator, but we use the kernel Ridge regression for $f(a, x)$. IPWCS is the IPW estimator $\hat{R}_{\text{IPWCS}}(\pi^e)$, where we use kernel regression defined in Section 5 to estimate $r(x)$ and $w(a, x)$. IPWCS-R is the same estimator, but we use KuLISF to estimate $r(x)$. Note that nuisance estimators in DM-R and IPWCS-R do not satisfy the Donsker condition.

The resulting MSE and the standard deviation (SD) over 20 replications of each experiment are shown in Tables 2, where we highlight in bold the best two estimators in each case. DRCS generally outperforms the other estimators. This result shows that the efficiency and double robustness of DRCS translate to satisfactory performance. IPW based estimators have unstable performance. While IPWCS-R shows the best performance in `satimage` dataset, it has severely low performance for `vehicle` dataset. IPWCS has a poor performance in both datasets. The larger instability of IPWCS-R is mainly due to the nuisance estimators in IPWCS-R do not satisfy the Donsker condition. When the behavior policy is similar to the evaluation policy, the DM estimators (DM and DM-R) also work well.

### 7.2 Experiments of Off-Policy Learning

For OPL, we compare the performances of three estimators of the optimal policy maximizing expected reward over the evaluation data: $\hat{\pi}_{\text{DRCS}}$ with $f(a, x)$ and $w(a, x)$ estimated by kernel Ridge regression and $r(x)$ estimated by KuLISF (DRCS), $\hat{\pi}_{\text{DM}}$ with $f(a, x)$ estimated by kernel regression defined in Section 5 (DM), and $\hat{\pi}_{\text{IPWCS}}$ with $r(x)$ and $w(a, x)$ estimated by kernel regression defined in Section 5 (IPWCS). For the policy class $\Pi$, we use a model with the Gaussian kernel defined in Appendix G. For DRCS, we use 2-fold cross-fitting and add a regularization term.

We conduct 10 trials for each experiment. The resulting expected reward over the evaluation data (RWD) and the standard deviation (SD) of estimators for OPL are shown in Table 3, where we highlight in bold the best estimator in each case. For all cases, the estimator $\hat{\pi}_{\text{DRCS}}$ outperforms the other estimators. We can find that, when an estimator of OPE shows high performance, a corresponding estimator of OPL also shows high performance. The results show that the statistical efficiency of the OPE estimator translates into better regret performance, as in Theorem 7.

## 8  Conclusion and Future Direction

We calculated the efficiency bound for OPE under a covariate shift and proposed OPE and OPL methods for the situation. In particular, DRCS has doubly robustness and achieves the efficiency bound under weak nonparametric rate conditions. The proposed OPE estimator is efficient under a simple setting in a transportability problem (Bareinboim & Pearl, 2016). Complete identification algorithms have been developed in a more complex setting (Bareinboim & Pearl, 2014); however, statistical efficient estimation methods have not been considered. Our work opens the door to this new direction. How to conduct efficient estimation in such a complex setting is an interesting future work.

## Broader Impact

Because the policies in sequential decision-making problems are critical in various real-world applications, the OPE methods are employed to evaluate the new policy and reduce the risk of deploying a poor policy. We focus on the OPE under a covariate shift between a historical and evaluation data. This setting has many practical applications. For example, in the advertising applications, we usually deliver advertisements only in the particular region to test the market in the beginning of the planned advertising campaign, then expand to other regions that have different feature distribution. Thus, we face the covariate shift in the evaluation and training a new policy for the new region.

Despite its practical importance, the OPE methods under the covariate shift have not been researched well, and people apply standard OPE methods to cases under the covariate shift. For instance, Hirano et al. (2003a) briefly discuss such a setting in Section 4.2, but did not discuss estimation of the density ratio, i.e., simply considered a case where the density ratio is known. As we explained, the standard methods are not robust against the covariate shift. Among the standard methods, the IPW estimator is not consistent, and the DM and DR estimator has consistency when the model of conditional outcome is correct. In particular, under a covariate shift, the standard DR estimator is *not* doubly robust; i.e., it is consistent only when the model of conditional outcome is correct. Thus, the standard estimator has a potential risk to mislead the user's decision making and might cause serious problems in the industry because many decision makings such as ad-optimization rely on the result of the evaluation. On the other hand, the proposed estimator is doubly robust. This robustness helps to avert the potential consequences of incorrect decision-making.

## Acknowledgement

Masatoshi Uehara was supported in part by MASASON Foundation.

## Footnotes

[2]We use $x$ and $z$ exchangeably noting that the spaces of $X$ and $Z$ are the same such as $q(x), q(z)$ and $p(x), p(z)$. On the other hand, we strictly distinguish $X_i$ and $Z_i$ noting that these are different random variables.

[3]Formally, regular estimators means estimators of which the limiting distribution is insensitive to local changes of the DGP. Refer to van der Vaart (1998, Chapter 7)

[4]When the square integrable envelope function exists and the metric entropy of the function class is controlled at some rates, the Donsker condition is satisfied (van der Vaart, 1998, Chapter 19).

[5]In this paragraph, we omit $\pi^{\mathrm{e}}$ from the estimator $\hat{R}(\pi^{\mathrm{e}})$.

[6] `https://www.csie.ntu.edu.tw/~cjlin/libsvmtools/datasets/multiclass.html`

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
