[Supplementary Material]

# A   Notations, Terms, and Abbreviations

In this section, we summarize the notations used in this paper.

Table 4: Summary of notations

| | |
|---|---|
| $A$, $X$, $Y$ | Action, covariate, reward |
| $\mathbb{E}[\mu(X, A, Y)]$ | $\mathbb{E}_{p(x)\pi^{\mathrm{b}}(a\mid x)p(y\mid a,x)}[\mu(x, a, y)]$ |
| $\mathbb{E}[\mu(Z)]$ | $\mathbb{E}_{q(z)}[\mu(z)]$ |
| $\pi^{\mathrm{b}}(a \mid x)$ | Behavior policy |
| $\pi^{\mathrm{e}}(a \mid x)$ | Evaluation policy |
| $R(\pi^{\mathrm{e}})$ | $\mathbb{E}_{q(x)\pi^{\mathrm{e}}(a\mid x)p(y\mid a,x)}[y]$ |
| $r(x)$ | $p(x)/q(x)$ |
| $w(a, x)$ | $\pi^{\mathrm{e}}(a \mid x)\pi^{\mathrm{b}}(a \mid x)$ |
| $f(a, x)$ | $\mathbb{E}_{p(y\mid a,x)}[y \mid a, x]$ |
| $v(x)$ | $\mathbb{E}_{\pi^{\mathrm{e}}(a\mid x)}[f(a, x) \mid x]$ |
| $\mathrm{Asmse}[\hat{R}]$ | $\lim_{n\to\infty} \mathbb{E}[(\hat{R} - R)^2]n$ |
| $\Pi$ | Policy class |
| $\otimes A$ | $AA^{\top}$ |
| $\|\mu(X)\|_2$, $\|\mu(X)\|_\infty$ | $L^2$-norm, $L^\infty$-norm |
| $k(\Pi)$ | Entropy integral of $\Pi$ w.r.t $\epsilon$-Hamming distance |
| $n^{\mathrm{hst}}$ | Number of training data |
| $n^{\mathrm{evl}}$ | Number of evaluation data |
| $A \lesssim B$ | There exists an absolute constant $C$ s.t. $A \le CB$ |
| $C_1, C_2, R_{\max}$ | Upper bound of $r(X), w(A, X), Y$ |
| $\rho$ | $n^{\mathrm{hst}}/(n^{\mathrm{hst}} + n^{\mathrm{evl}})$ |
| $\mathcal{D}^{\mathrm{hst}}$, $\mathcal{D}^{\mathrm{eval}}$ | Train data, evaluation data |
| $n_1^{\mathrm{hst}}, n_2^{\mathrm{hst}}$ | Split train data |
| $n_1^{\mathrm{evl}}, n_2^{\mathrm{evl}}$ | Split evaluation data |
| $\mathcal{D}_i$ | Concatenation of $n_i^{\mathrm{hst}}$ and $n_i^{\mathrm{evl}}$ $i = 1, 2$ |
| $\mathbb{G}_{n^{\mathrm{hst}}}$ | $\sqrt{n^{\mathrm{hst}}}\{\mathbb{E}_{n^{\mathrm{hst}}} - \mathbb{E}\}$ Empirical process based on train data |
| $\mathbb{G}_{n^{\mathrm{evl}}}$ | $\sqrt{n^{\mathrm{evl}}}\{\mathbb{E}_{n^{\mathrm{evl}}} - \mathbb{E}\}$ Empirical process based on evaluation data |
| $\Upsilon(\pi^{\mathrm{e}})$ | Semiaprametric lower bound of $R(\pi^{\theta})$ under nonparametric model |
| $K_h(\cdot)$ | Kernel with a bandwidth $h$ |
| $n_k^{\mathrm{hst}}$ | $k$-th train data |
| $n_k^{\mathrm{evl}}$ | $k$-th evaluation data |

# B   Identification under Potential Outcome Framework

We explain how to apply our results in the main draft under potential outcome framework, which is a common framework in causal inference literature (Rubin, 1987). In this section, our goal is justifying DM and IPWCS estimators under potential outcome framework.

Let us denote counterfactual variables based on stochastic policies (interventions) as $Y(B)$, where $B$ is a random variable from the conditional density $\pi^{\mathrm{e}}(b|Z)$ [7] and $Z$ is a random variable following the evaluation density $q(z)$. Here, note what we can observe is data:

$$\{X_i, A_i, Y_i\}_{i=1}^{n^{\mathrm{hst}}} \sim p(x)\pi^{\mathrm{e}}(a \mid x)p(y \mid a, x), \ \{Z_j\}_{j=1}^{n^{\mathrm{evl}}} \sim q(z).$$

A detailed review of the stochastic intervention is shown in Muñoz & Van Der Laan (2012); Young et al. (2014).

Then, let us put the following assumptions:

- Consistency: $Y = Y(a)$ if $A = a$ for $\forall a \in \mathcal{A}$,

- Unconfoundedness: $A$ and $Y(a)$ are conditionally independent given $X$ for any $a \in \mathcal{A}$, $G$ and $Y(a)$ are conditionally independent given $Z$ for any $a \in \mathcal{A}$,

- Transportability: $\mathrm{E}[Y(a) \mid Z = c] = \mathrm{E}[Y(a) \mid X = c]$ for any $a \in \mathcal{A}, c \in \mathcal{X}$.

Note that transportability is a weaker assumption compared with the assumption in the main draft:

$$p_{\text{train}}(Y(a) \mid c) = p_{\text{test}}(Y(a) \mid c),$$

where $p_{\text{train}}(\cdot \mid \cdot)$ is a condition density of $Y(a)$ given $Z$, $p_{\text{test}}(\cdot \mid \cdot)$ is a condition density of $Y(a)$ given $X$. Following Lemma 1 (Kennedy, 2019), we can prove the following lemma.

**Lemma 2** (G-formula). $\mathbb{E}[Y(B)] = \int \mathbb{E}[Y \mid A = a, X = x]\pi^{\mathrm{e}}(a \mid x)q(x)\mathrm{d}(a,x)$.

*Proof.*

$$
\begin{aligned}
\mathbb{E}[Y(B)] &= \int \mathbb{E}[Y(b) \mid B = b, Z = z]\pi^{\mathrm{e}}(b \mid z)q(z)\mathrm{d}(b,z) \\
&= \int \mathbb{E}[Y(g) \mid Z = z]\pi^{\mathrm{e}}(b \mid z)q(z)\mathrm{d}(b,z) \\
&= \int \mathbb{E}[Y(g) \mid X = z]\pi^{\mathrm{e}}(b \mid z)q(z)\mathrm{d}(b,z) \\
&= \int \mathbb{E}[Y(g) \mid A = g, X = z]\pi^{\mathrm{e}}(b \mid z)q(z)\mathrm{d}(b,z) \\
&= \int \mathbb{E}[Y(a) \mid A = a, X = x]\pi^{\mathrm{e}}(a \mid x)q(x)\mathrm{d}(a,x) \\
&= \int \mathbb{E}[Y \mid A = a, X = x]\pi^{\mathrm{e}}(a \mid x)q(x)\mathrm{d}(a,x).
\end{aligned}
$$

From the first line to the second line, we use a uncounfedness assumption. From the second line to the third line, we use a transportability assumption. From the third line to the fourth line, we use a uncounfedness assumption. From the fourth line to the fifth line, the random variables $a, x$ are replaced with $b, z$. From the fifth line to the sixth line, we use a consistency assumption.

□

From this lemma, the DM method can be naturally introduced. Note this is equivalent to a transport formula Pearl & Bareinboim (2014, (3.1)) when the evaluation policy is atomic. The G-formula described here is its extension when the evaluation policy is stochastic.

**Theorem 8** (IPWCS). $\mathbb{E}[Y(B)] = \mathbb{E}[r(X)w(A, X)Y]$

*Proof.*

$$
\begin{aligned}
\mathbb{E}[r(X)w(A, X)Y] &= \mathbb{E}[r(X)w(A, X)\mathbb{E}[Y \mid A, X]] \\
&= \int \mathbb{E}[Y \mid A = a, X = x]r(x)w(a,x)\pi^{\mathrm{b}}(a \mid x)p(x)\mathrm{d}(a,x) \\
&= \int \mathbb{E}[Y \mid A = a, X = x]\pi^{\mathrm{e}}(a \mid x)q(x)\mathrm{d}(a,x) \\
&= \mathbb{E}[Y(B)].
\end{aligned}
$$

From the third line to the fourth line, we use a Lemma 2.

□

## C  Density Ratio Estimation

Here, we introduce the formulation of LSIF. In LSIF, we estimate the density ratio $r(x) = \frac{q(x)}{p(x)}$ directly. Let $\mathcal{S}$ be the class of non-negative measurable functions $s : \mathcal{X} \to \mathbb{R}^+$. We consider minimizing the following squared error between $s$ and $r$:

$$\mathbb{E}_{p(x)}[(s(x) - r(x))^2] = \mathbb{E}_{p(x)}[(r(x))^2] - 2\mathbb{E}_{q(z)}[s(z)] + \mathbb{E}_{p(x)}[(s(x))^2]. \tag{7}$$

The first term of the last equation does not affect the result of minimization and we can ignore the term, i.e., the density ratio is estimated through the following minimization problem:

$$s^* = \arg\min_{s \in \mathcal{S}} \left[ \frac{1}{2} \mathbb{E}_{p(x)}[(s(x))^2] - \mathbb{E}_{q(z)}[s(z)] \right],$$

where $\mathcal{S}$ is a hypothesis class of the density ratio. As mentioned above, to minimize the empirical version of (7), we use uLSIF (Sugiyama et al., 2012). Given a hypothesis class $\mathcal{H}$, we obtain $\hat{r}$ by

$$\hat{r} = \arg\min_{s \in \mathcal{H}} \left[ \frac{1}{2} \mathbb{E}_{n^{\text{hst}}}[(s(X))^2] - \mathbb{E}_{n^{\text{evl}}}[s(Z)] + \mathcal{R}(s) \right], \text{ where } \mathcal{R} \text{ is a regularization term. For}$$

a model of uLSIF, Kanamori et al. (2012) proposed using kernel based hypothesis to estimate the density ratio nonparametrically. Kanamori et al. (2012) called uLSIF with kernel based hypothesis as KuLSIF. Kanamori et al. (2012) showed that, under some assumptions, the convergence rate of KuLSIF is $\left\| \hat{r}(X) - \left( \frac{q(X)}{p(X)} \right) \right\|_2 = O_p \left( \min \left( n^{\text{hst}}, n^{\text{evl}} \right)^{-\frac{1}{2+\gamma}} \right)$, where $0 < \gamma < 2$ is a constant depending on the bracketing entropy of $\mathcal{H}$.

## D   Efficiency bound for the stratified sampling mechanism

In this section, we discuss the efficiency bound.

### D.1   Cramér-Rao lower bound

First, we show the Cramér-Rao lower bound when the DGP is a stratified sampling with the historical data $\{\alpha_i\}_{i=1}^{n^{\text{hst}}}$ and evaluation data $\{\beta_i\}_{i=1}^{n^{\text{evl}}}$, where $\alpha_i$ and $\beta_i$ are random variables. Let $H_{n^{\text{hst}}}$ and $G_{n^{\text{evl}}}$ be the distributions of $\{\alpha_i\}_{i=1}^{n^{\text{hst}}}$ and $\{\beta_i\}_{i=1}^{n^{\text{evl}}}$. Let us define a set of distributions as $\mathcal{M}_n = \{H_{n^{\text{hst}}}, G_{n^{\text{evl}}}\}$. A model $\mathcal{M}_n^{\text{para}}$ is called a regular parametric submodel if the model can be written as $\mathcal{M}_n^{\text{para}} = \{H_{\theta_1, n^{\text{hst}}}, G_{\theta_2, n^{\text{evl}}}\}$, where $\theta_1 \in \Theta_1$, $\theta_2 \in \Theta_2$ and it matches the true distribution at $\theta_1^*$ and $\theta_2^*$, and it has a density

$$h_{\theta_1, n^{\text{hst}}}(\{\alpha_i\}) = \prod_{i=1}^{n^{\text{hst}}} h(\alpha_i; \theta_1), \ g_{\theta_2, n^{\text{evl}}}(\{\beta_i\}) = \prod_{i=1}^{n^{\text{evl}}} g(\beta_i; \theta_2).$$

Let $R(H, G) \to \mathbb{R}$ be a target functional. Then, the Cramér-Rao lower bound of the functional $R$ under the parametric submodel $\mathcal{M}_n^{\text{para}}$ is

$$\begin{aligned} \text{CR}(\mathcal{M}_n^{\text{para}}, R) =& \nabla_{\theta_1^\top} R(H_{\theta_1}, G_{\theta_2}) \mathbb{E}[\otimes \nabla_{\theta_1} \log h_{\theta_1, n^{\text{hst}}}]^{-1} \nabla_{\theta_1} R(H_{\theta_1}, G_{\theta_2}) \\ &+ \nabla_{\theta_2^\top} R(H_{\theta_1}, G_{\theta_2}) \mathbb{E}[\otimes \nabla_{\theta_1} \log g_{\theta_2, n^{\text{evl}}}]^{-1} \nabla_{\theta_2} R(H_{\theta_1}, G_{\theta_2}). \end{aligned}$$

Before that, we calculate the Cramér-Rao lower bound in a tabular setting, where the state, action and reward spaces are finite.

**Theorem 9.** *In a tabular case, $n\text{CR}(\mathcal{M}_n^{\text{para}}, R)$ is*

$$\rho^{-1} \mathbb{E}[r^2(X) w^2(A, X) \text{var}[Y \mid A, X]] + (1 - \rho)^{-1} \text{var}[v(Z)]$$

.

*Proof of Theorem 9.* In our setting, we have $\{X_i, A_i, Y_i\}_{i=1}^{n^{\text{hst}}}$ and $\{Z_j\}_{j=1}^{n^{\text{evl}}}$. The target functional, i.e., the value of the evaluation policy $\pi^{\text{e}}$ defined in (1) is

$$R(\pi^{\text{e}}) = \int y q(x) \pi^{\text{e}}(a \mid x) p(y \mid a, x) \mathrm{d}\mu(a, x, y), \tag{8}$$

where $\mu$ is a baseline measure such as Lebesgue or counting measure. The scaled Cramér-Rao lower bound for regular parametric models under $\mathcal{M}_n^{\text{para}}$:

$$n\text{CR}(\mathcal{M}_n^{\text{para}}, R)$$

is given by

$$\rho^{-1} A_1 B_1^{-1} A_1^\top + (1 - \rho)^{-1} A_2 B_2^{-1} A_2^\top \tag{9}$$

$$A_1 = \mathbb{E}_{x \sim q(x),\, a \sim \pi^{\text{e}}(a|x),\, y \sim p(y|a,x)}[y \nabla_{\theta_1^\top} \log p(y \mid a, x; \theta_1)],$$

$$A_2 = \mathbb{E}_{x \sim q(x),\, a \sim \pi^{\text{e}}(a|x),\, y \sim p(y|a,x)}[y \nabla_{\theta_2^\top} \log q(x; \theta_2)],$$

$$B_1 = \mathbb{E}_{x \sim p(x), a \sim \pi^{\text{b}}(a|x), y \sim p(y|a,x)}[\otimes \nabla_{\theta_1} \log p(y \mid a, x; \theta_1)],$$

$$B_2 = \mathbb{E}_{z \sim q(z)}[\otimes \nabla_{\theta_2} \log q(z; \theta_2)].$$

Then, from the Cauchy Schwartz inequality (Tripathi, 1999), we have the following inequality:
$$\mathbb{E}[A(Z)B^\top(Z)]\mathbb{E}[B(Z)B^\top(Z)]^{-1}\mathbb{E}[A(Z)B^\top(Z)]^\top \leq \mathbb{E}[A^2(Z)],$$
where $\mathbb{E}[A(Z)]=0, \mathbb{E}[B(Z)]=0$. Then, we obtain the following upper bound:

$$
\begin{aligned}
&A_1 B_1^{-1} A_1^\top \\
&= \mathbb{E}[r(X)w(A,X)Y\nabla_{\theta_1^\top}\log p(Y\mid A,X;\theta_1)]\mathbb{E}[\otimes\nabla_{\theta_1}\log p(Y\mid A,X;\theta_1)]^{-1}\\
&\quad \times \mathbb{E}[r(X)w(A,X)Y\nabla_{\theta_1}\log p(Y\mid A,X;\theta_1)]\\
&= \mathbb{E}[r(x)w(A,X)\{Y-\mathbb{E}[Y\mid A,X]\}\nabla_{\theta_1^\top}\log p(Y\mid A,X;\theta_1)]\mathbb{E}[\otimes\nabla_{\theta_1}\log p(Y\mid A,X;\theta_1)]^{-1}\\
&\quad \times \mathbb{E}[r(x)w(A,X)\{Y-\mathbb{E}[Y\mid A,X]\}\nabla_{\theta_1}\log p(Y\mid A,X;\theta_1)]\\
&\leq \mathbb{E}[r^2(X)w^2(A,X)\{Y-\mathbb{E}[Y\mid A,X]\}^2] = \mathbb{E}[r^2(X)w^2(A,X)\mathrm{var}[Y\mid A,X]].
\end{aligned}
$$

In the same way,

$$
\begin{aligned}
&A_2 B_1^{-1} A_2^\top\\
&= \mathbb{E}[v(Z)\nabla_{\theta_2^\top}\log q(Z;\theta_2)]\mathbb{E}[\otimes\nabla_{\theta_2}\log q(Z;\theta_2)]^{-1}\mathbb{E}[v(Z)\nabla_{\theta_2}\log q(z;\theta_2)]\\
&= \mathbb{E}[\{v(Z)-\mathbb{E}[v(Z)]\}\nabla_{\theta_2^\top}\log q(Z;\theta_2)]\mathbb{E}[\otimes\nabla_{\theta_2}\log q(Z;\theta_2)]^{-1}\mathbb{E}[\{v(Z)-\mathbb{E}[v(Z)]\}\nabla_{\theta_2}\log g(Z;\theta_2)]\\
&\leq \mathbb{E}[\{v(Z)-\mathbb{E}[v(Z)]\}^2] = \mathrm{var}[v(Z)].
\end{aligned}
$$

Therefore,
$$
\begin{aligned}
&\rho^{-1}A_1 B_1^{-1} A_1^\top + (1-\rho)^{-1}A_2 B_2^{-1} A_2^\top\\
&\leq \rho^{-1}\mathbb{E}[r^2(X)w^2(A,X)\mathrm{var}[Y\mid A,X]] + (1-\rho)^{-1}\mathrm{var}[v(Z)].
\end{aligned}
$$

Finally, we have to show this inequality is equality. This is obvious because our setting is tabular. □

## D.2 Reduction to i.i.d setting

The Cramér-Rao lower bound we have seen so far can be extended when the model is semiparametric. However, because our DGP is not i.i.d, we cannot direct apply standard semiparametric theory here. To circumvent this problem, we regard the whole $n$ data at hand as one observation and consider the case where we observe $m$ observations. Then, as $m$ goes to infinity, the total data size $n' := nm$ goes to infinity. Because each one observation ($n$ data) is i.i.d, we can apply standard semiparametric theory. Only in this section, we regard $n'$ as the total hypothetical sample size when discussing asymptotics. The value $n$ is the sample size of the actual data at hand, which is fixed.

We explain the definition of the efficient influence function (EIF). This is a function for one observation
$$o = (x_1,\cdots,x_{n^{\mathrm{hst}}}, a_1,\cdots,a_{n^{\mathrm{hst}}}, y_1,\cdots,y_{n^{\mathrm{hst}}}, z_1,\cdots,z_{n^{\mathrm{evl}}}).$$
This is defined given the target functional and the model. In our context, the EIF has the following property.

**Theorem 10.** *(van der Vaart, 1998, Theorem 25.20) The EIF $\phi(o)$ is the gradient of $R(\pi^{\mathrm{e}})$ w.r.t the model $\mathcal{M}_n$, which has the smallest $L_2$-norm. It satisfies that for any regular estimator $\hat{R}$ of $R(\pi^{\mathrm{e}})$ w.r.t the model $\mathcal{M}_n$, $\mathrm{AMSE}[\hat{R}] \geq \mathrm{var}[\phi(o)]$, where $\mathrm{AMSE}[\hat{R}]$ is the second moment of the limiting distribution of $\sqrt{m}(\hat{R}-R(\pi^{\mathrm{e}}))$.*

This states that $\mathrm{var}[\phi(o)]$ is the lower bound in estimating $R(\pi^{\mathrm{e}})$. We call $n\mathrm{var}[\phi(o)]$ the efficiency bound because what we want to consider is the lower bound of $\sqrt{n'}(\hat{R}-R(\pi^{\mathrm{e}}))$. Note that $n$ is fixed here.

For the current case, the EIF and efficiency bound are explicitly calculated as follows.

**Theorem 11.** *The EIF of $R(\pi^{\mathrm{e}})$ w.r.t the model $\mathcal{M}_n$ is*

$$\phi(o) = \frac{1}{n^{\mathrm{hst}}}\sum_{i=1}^{n^{\mathrm{hst}}} r(x_i)w(a_i,x_i)\{y_i - q(x_i,a_i)\} + \frac{1}{n^{\mathrm{evl}}}\sum_{j=1}^{n^{\mathrm{evl}}} v(z_j) - R(\pi)$$

*The (scaled) efficiency bound $n\mathrm{var}[\phi(o)]$ is*

$$\rho^{-1}\mathbb{E}[r^2(X)w^2(A,X)\mathrm{var}[Y\mid A,X]] + (1-\rho)^{-1}\mathrm{var}[v(Z)].$$

When assuming the model $\mathcal{M}_n^{\mathrm{fix}}$ where $\pi^{\mathrm{b}}(a|x)$ and $p(x)$ are fixed at true values, we can also show that the EIF and the efficiency bound are the same.

*Proof of Theorem 1.* We follow the following steps.

1. Calculate some gradient (a candidate of EIF) of the target functional $R(\pi^{\mathrm{e}})$ w.r.t $\mathcal{M}_n$.

2. Calculate the tangent space w.r.t $\mathcal{M}_n$.

3. Show that the candidate of EIF in Step 1 lies in the tangent space. Then, this concludes that a candidate of EIF in Step 1 is actually the EIF.

**Calculation of the gradient** As mentioned, the model $\mathcal{M}_n^{\mathrm{para}}$ for a nonparametric model $\mathcal{M}_n$ is

$$p(o; \theta) = \prod_{i=1}^{n^{\mathrm{hst}}} p(x_i; \theta_x) \pi^{\mathrm{b}}(a_i \mid x_i; \theta_a) p(y_i \mid x_i, a_i; \theta_y) \prod_{j=1}^{n^{\mathrm{evl}}} q(z_j; \theta_z),$$

$$\theta = (\theta_x^\top, \theta_a^\top, \theta_y^\top, \theta_z^\top)^\top, \ o = \{x_i, a_i, y_i, z_j\}_{i=1, j=1}^{n^{\mathrm{hst}}, n^{\mathrm{evl}}}.$$

We define the corresponding gradients:

$$g_x = \nabla_{\theta_x} \log p(x; \theta_x), \ g_{a|x} = \nabla_{\theta_a} \log \pi^{\mathrm{b}}(a|x; \theta_a), \ g_{y|a,x} = \nabla_{\theta_y} \log p(y|a, x; \theta_y), q_z = \nabla_{\theta_z} \log q(z; \theta_z).$$

To derive some gradient of the target functional $R(\pi^{\mathrm{e}})$ w.r.t $\mathcal{M}_n$, what we need is finding a function $f(o)$ satisfying

$$\nabla R(\theta) = \mathbb{E}[f(\mathcal{D}) \nabla \log p(\mathcal{D}; \theta)]$$

$$= \mathbb{E}\left[ f(\mathcal{D}) \left\{ \frac{1}{n^{\mathrm{hst}}} \sum_{i=1}^{n^{\mathrm{hst}}} \{g_x(X_i) + g_{a|x}(X_i, A_i) + g_{y|x,a}(X_i, A_i, Y_i)\} + \frac{1}{n^{\mathrm{evl}}} \sum_{j=1}^{n^{\mathrm{evl}}} g_z(Z_j) \right\} \right].$$

We take the derivative as follows:

$$\nabla R(\theta) = \mathbb{E}_{q(x)\pi^{\mathrm{e}}(a|x)p(y|a,x)} \left[ y \left\{ g_z(x) + g_{y|a,x}(y|a, x) \right\} \right].$$

By some algebra, this is equal to

$$\mathbb{E}\left[ \left\{ \frac{1}{n^{\mathrm{hst}}} \sum_{i=1}^{n^{\mathrm{hst}}} r(X_i) w(A_i, X_i) \{Y_i - q(X_i, A_i)\} + \frac{1}{n^{\mathrm{evl}}} \sum_{j=1}^{n^{\mathrm{evl}}} v(Z_j) - R(\pi) \right\} \nabla \log p(\mathcal{D}; \theta) \right].$$

Thus, the following function

$$\phi(o) = \frac{1}{n^{\mathrm{hst}}} \sum_{i=1}^{n^{\mathrm{hst}}} r(x_i) w(a_i, x_i) \{y_i - q(x_i, a_i)\} + \frac{1}{n^{\mathrm{evl}}} \sum_{j=1}^{n^{\mathrm{evl}}} v(z_j) - R(\pi)$$

is a derivative.

**Calculation of the tangent space** Following a standard derivation way (Tsiatis, 2006; van der Vaart, 1998), the tangent space of the model $\mathcal{M}_n$ is

$$\left\{ \frac{1}{n^{\mathrm{hst}}} \sum_{i=1}^{n^{\mathrm{hst}}} \{t_x(x_i) + t_{a|x}(x_i, a_i) + t_{y|a,x}(x_i, a_i, y_i)\} + \frac{1}{n^{\mathrm{evl}}} \sum_{j=1}^{n^{\mathrm{evl}}} t_z(z_j) \in L_2(o) \right\}.$$

where $L_2(o)$ is an $L_2$ space at the true density,

$$\mathbb{E}[t_x(X)] = 0, \mathbb{E}[t_{a|x}(X, A)|X] = 0, \mathbb{E}[t_{y|a,x}(X, A, Y)|X, A] = 0, \mathbb{E}[t_z(Z)] = 0.$$

**Last Part** We can easily check that $\phi(o)$ lies in the tangent space by taking

$$t_x = 0, t_{a|x} = 0, t_{y|a,x} = r(x)w(a,x)\{y - q(a,x)\}, t_z(z) = v(z) - R(\pi).$$

Thus, $\phi(o)$ is the EIF.

**Remark 6.** We can easily see that the EIF is $\phi(o)$ when assuming the model $\mathcal{M}_n^{\text{fix}}$ where $p(x)$ and $\pi^{\text{b}}(a|x)$ are fixed at true values. This model is represented as

$$\left\{ \textstyle\prod_{i=1}^{n^{\text{hst}}} p_*(x_i)\pi_*^{\text{b}}(a_i \mid x_i)p(y_i \mid x_i, a_i; \theta_y) \prod_{j=1}^{n^{\text{evl}}} q(z_j; \theta_z) \right\}$$

where $\cdot_*$ emphasizes that these are fixed at true densities.

The function $\phi(o)$ in the proof is still a gradient of $R(\pi^{\text{e}})$ w.r.t $\mathcal{M}_n^{\text{fix}}$ because the model $\mathcal{M}_n^{\text{fix}}$ is smaller than the model $\mathcal{M}_n$. Besides, $\phi(o)$ belongs to the tangent space induced by the model $\mathcal{M}_n^{\text{fix}}$ because the tangent space induced by $\mathcal{M}_n^{\text{fix}}$ is

$$\left\{ \frac{1}{n^{\text{hst}}} \sum_{i=1}^{n^{\text{hst}}} \{t_{y|a,x}(x_i, a_i, y_i)\} + \frac{1}{n^{\text{evl}}} \sum_{j=1}^{n^{\text{evl}}} t_z(z_j) \in L_2(o) \right\}$$

where

$$\mathbb{E}[t_{y|a,x}(X, A, Y)|X, A] = 0, \mathbb{E}[t_z(Z)] = 0.$$

We can easily check that $\phi(o)$ lies in the above tangent space by taking

$$t_{y|a,x} = r(x)w(a,x)\{y - q(a,x)\}, t_z(z) = v(z) - R(\pi).$$

Thus, $\phi(o)$ is the EIF regarding $\mathcal{M}_n^{\text{fix}}$. □

# E   Proofs

In this section, we show the proofs of theorems. In the proofs of Theorem 2–14, we prove the case where we use a two-fold cross-fitting. The extension of two fold cross-fitting to the general $K$-fold cross-fold is straightforward.

## E.1   Required conditions

In order to show Theorems 6–13, we use the following Theorem 12, which shows the convergence rate of kernel regression. Here, we have data $\{B_i, C_i\}_{i=1}^n$, which are i.i.d. from $p(b,c) = p(c|b)p(b)$, and $B_i$ takes a value in $\mathcal{B}$. Then, let us consider a kernel estimation:

$$n^{-1} \textstyle\sum_{i=1}^n K_h(B_i - b)C_i,$$

where $K_h(b) = h^{-d}K(b/h^d)$, where $d$ is a dimension of $b$. Then, we have the following theorem following Newey & Mcfadden (1994).

**Theorem 12.** *Assume*

- *the space $\mathcal{B}$ is compact and $p(b) > 0$ on $\mathcal{B}$,*

- *the kernel $K(u)$ has the bounded derivative of order $k$, satisfies $\int K(u)\mathrm{d}u = 1$, and has zero moments of order $\leq m - 1$ and a nonzero $m$-th order moment,*

- *$\mathbb{E}[C \mid B = b]$ is continuously differentiable to order $k$ with bounded derivatives on the opening set in $\mathcal{B}$.*

- *there is $v \geq 4$ such that $\mathrm{E}[|C|^v] \leq \infty$ and $\mathrm{E}[|C|^v \mid B = b]p(b)$ is bounded.*

*Then, when $h = h(n)$ and $h(n) \to 0$,*

$$\|n^{-1} \textstyle\sum_{i=1}^n K_h(B_i - b)C_i - p(b)\mathbb{E}[C|b]\|_\infty = \mathrm{O}_p\left(\frac{\log n^{1/2}}{(nh^{d+2k})^{1/2}} + h^m\right). \tag{10}$$

*Then, under $n^{1-2/v}h^d/\log n \to \infty$, $\sqrt{n}h^{d+2k} \to \infty$, $\sqrt{n}h^{2m} \to 0$, the above $l_\infty$ risk is $\mathrm{o}_p(n^{-1/2})$ (Newey & Mcfadden, 1994).*

**Additional assumptions:** regarding Theorem 12, we assume the following assumptions when we prove Theorems 13–6:

**Theorem 13** : condition when replacing $B$ with $X$ , $C$ with $w(A, X)Y$, condition when replacing $B$ with $Z$, $C$ with 1.

**Theorem 4** : condition when replacing $B$ with $X$ , $C$ with $w(A, X)Y$, condition when replacing $B$ with $X$, $C$ with 1, condition when replacing $B$ with $Z$, $C$ with 1.

**Thorem 5** : condition when replacing $B$ with $(X, A)$, $C$ with 1, condition when replacing $B$ with $(X, A)$, $C$ with $Y$, condition when replacing $B$ with $X$, $C$ with $w(A, X)Y$, and condition when replacing $B$ with $Z$, $C$ with 1

**Theorem 6** : condition when replacing $B$ with $(X, A)$, $C$ with $Y$, condition when replacing $B$ with $(X, A)$, $C$ with 1

## E.2    Warming up

As a warm up, first, we prove the asymptotic property of some simple estimator. When $p(x)$ and $\pi^{\mathrm{b}}(a \mid x)$ are known, let us define an IPW estimator:

$$\hat{R}_{\mathrm{IPW1}}(\pi^{\mathrm{e}}) = \mathbb{E}_{n^{\mathrm{hst}}} \left[ \frac{\hat{q}(X)}{p(X)} \frac{\pi^{\mathrm{e}}(A \mid X)Y}{\pi^{\mathrm{b}}(A \mid X)} \right].$$

**Theorem 13.** *When $\hat{q}(x) = \hat{q}_h(x)$, the asymptotic MSE of $\hat{R}_{\mathrm{IPW1}}$ is*

$$\rho^{-1}\mathrm{var}[r(X)w(A, X)Y] + (1 - \rho)^{-1}\mathrm{var}[v(Z)].$$

*Proof of Theorem 13.* We follow the proof of Newey & Mcfadden (1994). For the ease of notation, assume $\rho = k_1/(k_1 + k_2)$. In this case, $n^{\mathrm{hst}} = k_1 N_o$ and $n^{\mathrm{evl}} = k_2 N_o$, where $N_o = n/(k_1 + k_2)$. Note that in this asymptotic regime, $N_o \to \infty$. Therefore, we reindex the sample set as

$$\{X_i\}_{i=1}^{n^{\mathrm{hst}}} = \{X_{b,i}\}\,(1 \le b \le k_1,\, 1 \le i \le N_o),$$
$$\{Z_i\}_{j=1}^{n^{\mathrm{hst}}} = \{Z_{c,j}\}\,(1 \le c \le k_2,\, 1 \le j \le N_o).$$

Here, we only consider the estimator $\hat{R}_{\mathrm{IPW1}}(\pi^{\mathrm{e}})$ based on based on $\{X_{b,i}\}_{i=1}^{N_o}$ and $\{Z_{c,j}\}_{j=1}^{N_o}$, and denote it as $\hat{R}_{b,c}$. Then, the final estimator $\hat{R}_{\mathrm{IPW1}}(\pi^{\mathrm{e}})$ using all set of samples is equal to

$$\frac{1}{k_1 k_2} \sum_{b=1}^{k_1} \sum_{c=1}^{k_2} \hat{R}_{b,c},$$

because the kernel estimator has a linear property. More specifically, we have

$$
\begin{aligned}
\hat{R}_{\mathrm{IPW1}}(\pi^{\mathrm{e}}) &= \frac{1}{n^{\mathrm{hst}}} \sum_{i=1}^{n^{\mathrm{hst}}} \left\{ \frac{1}{n^{\mathrm{evl}}} \sum_{j=1}^{n^{\mathrm{evl}}} K_h(Z_j - X_i) \right\} \frac{\pi^{\mathrm{e}}(A_i \mid X_i)Y_i}{\pi^{\mathrm{b}}(A_i \mid X_i)p(X_i)} \\
&= \frac{1}{n^{\mathrm{hst}} n^{\mathrm{evl}}} \sum_{i=1}^{n^{\mathrm{hst}}} \sum_{j=1}^{n^{\mathrm{evl}}} \frac{K_h(Z_j - X_i)\pi^{\mathrm{e}}(A_i \mid X_i)Y_i}{\pi^{\mathrm{b}}(A_i \mid X_i)p(X_i)} \\
&= \frac{1}{k_1 k_2} \sum_{b=1}^{k_1} \sum_{c=1}^{k_2} \left\{ \frac{1}{n^{\mathrm{hst}} n^{\mathrm{evl}}} \sum_{i=1}^{n^{\mathrm{hst}}} \sum_{j=1}^{n^{\mathrm{evl}}} \frac{K_h(Z_{c,j} - X_{b,i})\pi^{\mathrm{e}}(A_{b,i} \mid X_{b,i})Y_{b,i}}{\pi^{\mathrm{b}}(A_{b,i} \mid X_{b,i})p(X_{b,i})} \right\} \\
&= \frac{1}{k_1 k_2} \sum_{b=1}^{k_1} \sum_{c=1}^{k_2} \hat{R}_{b,c}.
\end{aligned}
$$

First, we analyze $\hat{R}_{1,1}$.

**Step 1**   We prove the following in this step:

$$\hat{R}_{b,c} = \frac{1}{N_o}\sum_{i=1}^{N_o} r(X_{b,i})w(X_{b,i}, A_{b,i})Y_{b,i} + \frac{1}{N_o}\sum_{j=1}^{N_o} v(Z_{c,j}) + \mathrm{o}_p(n^{-1/2}).$$

Especially, we prove the statement for $\hat{R}_{1,1}$ when $k_1 = 1, k_2 = 1, n^{\mathrm{hst}} = n^{\mathrm{evl}} = n/2$. We have

$$\hat{R}_{1,1} = \mathbb{E}_{n^{\mathrm{hst}}}\left[\frac{\hat{q}_h(X)}{p(X)}\frac{\pi^{\mathrm{e}}(A \mid X)Y}{\pi^{\mathrm{b}}(A \mid X)}\right]$$

$$= \frac{1}{n^{\mathrm{hst}}}\sum_{i=1}^{n^{\mathrm{hst}}}\frac{1}{p(X_i)}\frac{\pi^{\mathrm{b}}(A_i \mid X_i)Y_i}{\pi^{\mathrm{b}}(A_i \mid X_i)}\left\{\frac{1}{n^{\mathrm{evl}}}\sum_{j=1}^{n^{\mathrm{evl}}} K_h(Z_j - X_i)\right\}$$

$$= \mathbb{E}_{n^{\mathrm{hst}}}\left[\frac{q(X)}{p(X)}\frac{\pi^{\mathrm{e}}(A \mid X)Y}{\pi^{\mathrm{b}}(A \mid X)}\right] + \frac{1}{n^{\mathrm{hst}}n^{\mathrm{evl}}}\sum_{i=1}^{n^{\mathrm{hst}}}\sum_{j=1}^{n^{\mathrm{evl}}} a_{i,j}$$

$$= \mathbb{E}_{n^{\mathrm{hst}}}\left[\frac{q(X)}{p(X)}\frac{\pi^{\mathrm{e}}(A \mid X)Y}{\pi^{\mathrm{b}}(A \mid X)}\right] + \frac{2}{n^{\mathrm{hst}}n^{\mathrm{evl}}}\sum_{i<j} b_{i,j},$$

where

$$a_{i,j}((X_i, A_i, Y_i), (Z_j)) = \frac{1}{p(X_i)}\frac{\pi^{\mathrm{b}}(A_i \mid X_i)Y_i}{\pi^{\mathrm{b}}(A_i \mid X_i)}\{K_h(Z_j - X_i) - q(X_i)\},$$

$$b_{i,j}((X_i, A_i, Y_i, Z_i), (X_j, A_j, Y_j, Z_j)) = 0.5\{a_{i,j} + a_{j,i}\}.$$

Then,

$$\frac{2}{n^{\mathrm{hst}}n^{\mathrm{evl}}}\sum_{i<j} b_{i,j}(X_i, A_i, Y_i, Z_i), (X_j, A_j, Y_j, Z_j))$$

$$= \frac{2}{n^{\mathrm{hst}}}\left\{\sum_{i=1}^{n^{\mathrm{hst}}}\mathbb{E}[b_{i,j} \mid X_i, A_i, Y_i, Z_i]\right\} + \mathrm{o}_p(n^{-1/2})$$

$$= \frac{1}{n^{\mathrm{evl}}}\sum_{i=1}^{n^{\mathrm{evl}}}\mathbb{E}[a_{j,i} \mid X_i, A_i, Y_i, Z_i] + \frac{1}{n^{\mathrm{hst}}}\sum_{i=1}^{n^{\mathrm{hst}}}\mathbb{E}[a_{i,j} \mid X_i, A_i, Y_i, Z_i] + \mathrm{o}_p(n^{-1/2})$$

$$= \frac{1}{n^{\mathrm{evl}}}\sum_{i=1}^{n^{\mathrm{evl}}}\{v(Z_i) - R(\pi^{\mathrm{e}})\} + \mathrm{o}_p(n^{-1/2}).$$

From the first line to the second line, we used the U-statistics theory (van der Vaart, 1998, Theorem 12.3). From the third line to the fourth line, based on Theorem 12, we used

$$\mathbb{E}[a_{j,i} \mid X_i, A_i, Y_i, Z_i] = \mathrm{o}_p(n^{-1/2}) + \mathbb{E}[w(A_i, X_i)Y_i \mid X_i = Z_i]\frac{p(X_i)}{p(X_i)} + \mathbb{E}[r(X_i)w(A_i, X_i)Y_i]$$

$$= \mathrm{o}_p(n^{-1/2}) + v(Z_i) - R(\pi^{\mathrm{e}}),$$

$$\mathbb{E}[a_{i,j} \mid X_i, A_i, Y_i, Z_i] = \mathrm{o}_p(n^{-1/2}) + \frac{1}{p(X_i)}\frac{\pi^{\mathrm{b}}(A_i \mid X_i)Y_i}{\pi^{\mathrm{b}}(A_i \mid X_i)}\{q(X_i) - q(X_i)\}$$

$$= \mathrm{o}_p(n^{-1/2}).$$

**Remark 7.** $\mathbb{E}[h(A_i, X_i, Y_i) \mid X_i = Z_i]$ is an abbreviation of $\{\mathbb{E}[h(A_i, X_i, Y_i) \mid X_i = x]\}_{x=Z_i}$.

Therefore,

$$\hat{R}_{\mathrm{IPWCSB}} = \mathbb{E}_{n^{\mathrm{hst}}}\left[\frac{q(X)}{p(X)}\frac{\pi^{\mathrm{e}}(A \mid X)Y}{\pi^{\mathrm{b}}(A \mid X)}\right] + \mathbb{E}_{n^{\mathrm{evl}}}[v(Z)] - R(\pi^{\mathrm{e}}) + \mathrm{o}_p(n^{-1/2}).$$

**Step 2**  Based on Step 1, we have

$$
\hat{R}_{\mathrm{IPW1}} = \frac{1}{k_1 k_2} \sum_{b=1}^{k_1} \sum_{c=1}^{k_2} \hat{R}_{b,c}
$$

$$
= \frac{1}{k_1 k_2} \sum_{b=1}^{k_1} \sum_{c=1}^{k_2} \left[ \frac{1}{N_o} \sum_{i=1}^{N_o} r(X_{b,i}) w(X_{b,i}, A_{b,i}) Y_{b,i} + \frac{1}{N_o} \sum_{j=1}^{N_o} \{ v(Z_{c,j}) \} \right] - R(\pi^{\mathrm{e}}) + \mathrm{o}_p(n^{-1/2})
$$

$$
= \frac{1}{k_1 N_o} \sum_{b=1}^{k_1} \left[ \sum_{i=1}^{N_o} r(X_{b,i}) w(X_{b,i}, A_{b,i}) Y_{b,i} \right] + \frac{1}{k_2 N_o} \sum_{c=1}^{k_2} \left[ \sum_{j=1}^{N_o} v(Z_{c,j}) \right] - R(\pi^{\mathrm{e}}) + \mathrm{o}_p(n^{-1/2})
$$

$$
= \frac{1}{n^{\mathrm{hst}}} \sum_{i=1}^{n^{\mathrm{hst}}} r(X_i) w(X_i, A_i) Y_i + \frac{1}{n^{\mathrm{evl}}} \sum_{j=1}^{n^{\mathrm{evl}}} v(Z_j) - R(\pi^{\mathrm{e}}) + \mathrm{o}_p(n^{-1/2}).
$$

Finally, from stratified sampling CLT, the statement is concluded. $\square$

### E.3  Proof of Theorem 2

*Proof.* We denote

$$
\phi_1(x, a, y; r, w, f) = r(x) w(a, x) \{ y - f(a, x) \}, \quad \phi_2(z; f) = v(z).
$$

We also denote the union of $n_i^{\mathrm{hst}}$ and $n_i^{\mathrm{evl}}$ as $\mathcal{D}_i$ for $i = 1, 2$, and the number of $n_1^{\mathrm{hst}}, n_2^{\mathrm{hst}}, n_1^{\mathrm{evl}}, n_2^{\mathrm{evl}}$ as $n_{11}, n_{21}, n_{12}, n_{22}$. For simplicity, we assume $n_{11} = n_{12}, n_{21} = n_{22}$.

Then, we have

$$
\sqrt{n} \{ \mathbb{E}_{n_1^{\mathrm{hst}}} [\phi_1(X, A, Y; \hat{r}^{(1)}, \hat{w}^{(1)}, \hat{f}^{(1)})] + \mathbb{E}_{n_1^{\mathrm{evl}}} [\phi_2(Z; \hat{f}^{(1)})] - R(\pi^{\mathrm{e}}) \}
$$

$$
= \sqrt{n} \left\{ \frac{1}{\sqrt{n_{11}}} \mathbb{G}_{n_1^{\mathrm{evl}}} [\phi_1(X, A, Y; \hat{r}^{(1)}, \hat{w}^{(1)}, \hat{f}^{(1)}) - \phi_1(X, A, Y; r, w, f)] + \frac{1}{\sqrt{n_{12}}} \mathbb{G}_{n_1^{\mathrm{evl}}} [\phi_2(Z; \hat{f}^{(1)}) - \phi_2(Z; f)] \right\} \tag{11}
$$

$$
+ \sqrt{n} \{ \mathbb{E}[\phi_1(X, A, Y; \hat{r}^{(1)}, \hat{w}^{(1)}, \hat{f}^{(1)}) \mid \hat{r}^{(1)}, \hat{w}^{(1)}, \hat{f}^{(1)}] + \mathbb{E}[\phi_2(Z; \hat{f}^{(1)}) \mid \hat{f}^{(1)}] \tag{12}
$$
$$
- \mathbb{E}[\phi_1(X, A, Y; r, w, f)] - \mathbb{E}[\phi_2(Z; f)] \}
$$
$$
+ \sqrt{n} \{ \mathbb{E}_{n_1^{\mathrm{hst}}} [\phi_1(X, A, Y; r, w, f)] + \mathbb{E}_{n_1^{\mathrm{evl}}} [\phi_2(Z; f)] - R(\pi^{\mathrm{e}}) \}. \tag{13}
$$

The term (11) is $\mathrm{o}_p(1)$ by Step 1. The term Eq. (12) is also $\mathrm{o}_p(1)$ by Step 2 as follows.

**Step 1:**  Eq. (11) is $\mathrm{o}_p(1)$.

If we can show that for any $\epsilon > 0$,

$$
\lim_{n \to \infty} P[| \sqrt{n} \{ \frac{1}{\sqrt{n_{11}}} \mathbb{G}_{n_1^{\mathrm{hst}}} [\phi_1(X, A, Y; \hat{r}^{(1)}, \hat{w}^{(1)}, \hat{f}^{(1)}) - \phi_1(X, A, Y; r, w, f)] \tag{14}
$$
$$
+ \frac{1}{\sqrt{n_{12}}} \mathbb{G}_{n_1^{\mathrm{evl}}} [\phi_2(Z; \hat{f}^{(1)}) - \phi_2(Z; f)] \} | > \epsilon \mid \mathcal{D}_2] = 0,
$$

then by the bounded convergence theorem, we would have

$$
\lim_{n \to \infty} P[| \sqrt{n} \{ \frac{1}{\sqrt{n_{11}}} \mathbb{G}_{n_1^{\mathrm{hst}}} [\phi_1(X, A, Y; \hat{r}^{(1)}, \hat{w}^{(1)}, \hat{f}^{(1)}) - \phi_1(X, A, Y; r, w, f)]
$$
$$
+ \frac{1}{\sqrt{n_{12}}} \mathbb{G}_{n_1^{\mathrm{evl}}} [\phi_2(Z; \hat{f}^{(1)}) - \phi_2(Z; f)] \} | > \epsilon] = 0,
$$

yielding the statement.

To show (14), we show that the conditional mean is $0$ and the conditional variance is $\mathrm{o}_p(1)$. Then, (14) is proved by the Chebyshev inequality following the proof of (Kallus & Uehara, 2020, Theorem

4). The conditional mean is

$$
\mathbb{E}[\sqrt{n}\{\frac{1}{\sqrt{n_{11}}}\mathbb{G}_{n_1^{\text{hst}}}[\phi_1(X,A,Y;\hat{r}^{(1)},\hat{w}^{(1)},\hat{f}^{(1)}) - \phi_1(X,A,Y;r,w,f)]
$$
$$
+ \frac{1}{\sqrt{n_{12}}}\mathbb{G}_{n_1^{\text{evl}}}[\phi_2(Z;\hat{f}^{(1)}) - \phi_2(Z;f)]\} \mid \mathcal{D}_2]
$$
$$
= \mathbb{E}[\sqrt{n}\{\frac{1}{\sqrt{n_{11}}}\mathbb{G}_{n_1^{\text{hst}}}[\phi_1(X,A,Y;\hat{r}^{(1)},\hat{w}^{(1)},\hat{f}^{(1)}) - \phi_1(X,A,Y;r,w,f)]
$$
$$
+ \frac{1}{\sqrt{n_{12}}}\mathbb{G}_{n_1^{\text{evl}}}[\phi_2(Z;\hat{f}^{(1)}) - \phi_2(Z;f)]\} \mid \mathcal{D}_2,\hat{r}^{(1)},\hat{w}^{(1)},\hat{f}^{(1)}]
$$
$$
= 0.
$$

Here, we used a cross-fitting construction. More specifically, regarding the second term, we have

$$
\mathbb{E}[\mathbb{E}_{n_1^{\text{evl}}}[\phi_2(Z;\hat{f}^{(1)}) - \phi_2(Z;f)] - \mathbb{E}[\phi_2(Z;\hat{f}^{(1)}) - \phi_2(Z;f)] \mid \mathcal{D}_2,\hat{r}^{(1)},\hat{w}^{(1)},\hat{f}^{(1)}]
$$
$$
= \mathbb{E}[\mathbb{E}_{n_1^{\text{evl}}}[\phi_2(Z;\hat{f}^{(1)}) - \phi_2(Z;f)] \mid \mathcal{D}_2,\hat{r}^{(1)},\hat{w}^{(1)},\hat{f}^{(1)}] - \mathbb{E}[\phi_2(Z;\hat{f}^{(1)}) - \phi_2(Z;f)] \mid \hat{f}^{(1)}]
$$
$$
= \mathbb{E}[\phi_2(Z;\hat{f}^{(1)}) - \phi_2(Z;f) \mid \hat{f}^{(1)}] - \mathbb{E}[\phi_2(Z;\hat{f}^{(1)}) - \phi_2(Z;f) \mid \hat{f}^{(1)}] = 0.
$$

The conditional variance is bounded as

$$
\text{var}[\sqrt{n}\{\frac{1}{\sqrt{n_{11}}}\mathbb{G}_{n_{11}}[\phi_1(X,A,Y;\hat{r}^{(1)},\hat{w}^{(1)},\hat{f}^{(1)}) - \phi_1(X,A,Y;r,w,f) \mid \mathcal{D}_2]
$$
$$
+ \frac{1}{\sqrt{n_{12}}}\mathbb{G}_{n_{12}}[\phi_2(Z;\hat{f}^{(1)}) - \phi_2(Z;f)]\} \mid \mathcal{D}_2]
$$
$$
= \frac{n}{n_{11}}\text{var}[\phi_1(X,A,Y;\hat{r}^{(1)},\hat{w}^{(1)},\hat{f}^{(1)}) - \phi_1(X,A,Y;r,w,f) \mid \mathcal{D}_2]
$$
$$
+ \frac{n}{n_{22}}\text{var}[\phi_2(Z;\hat{f}^{(1)}) - \phi_2(Z;f) \mid \mathcal{D}_2]
$$
$$
\leq \frac{n}{n_{11}}\mathbb{E}[\{\hat{r}^{(1)}(X)\hat{w}^{(1)}(A,X)(Y - \hat{f}^{(1)}(A,X)) - r(X)w(A,X)(Y - f(A,X))\}^2 \mid \mathcal{D}_2]
$$
$$
+ \frac{n}{n_{22}}\mathbb{E}[\{\hat{v}^{(1)}(Z) - v(Z)\}^2 \mid \mathcal{D}_2] = \text{o}_p(1) + \text{o}_p(1) = \text{o}_p(1).
$$

Here, we used

$$
\frac{n}{n_{11}}\mathbb{E}[\{\hat{r}^{(1)}(X)\hat{w}^{(1)}(A,X)(Y - \hat{f}^{(1)}(A,X)) - r(X)w(A,X)(Y - f(A,X))\}^2 \mid \mathcal{D}_2] = \text{o}_p(1).
\tag{15}
$$

and

$$
\mathbb{E}[\{\hat{v}^{(1)}(Z) - v(Z)\}^2 \mid \mathcal{D}_2] = \text{o}_p(1).
\tag{16}
$$

The first equation (15) is proved by

$$
\mathbb{E}[\{\hat{r}^{(1)}\hat{w}^{(1)}(Y - \hat{f}^{(1)}) - rw(Y - f)\}^2 \mid \mathcal{D}_2]
$$
$$
= \mathbb{E}[\{\hat{r}^{(1)}\hat{w}^{(1)}(Y - \hat{f}^{(1)}) - \hat{r}^{(1)}\hat{w}^{(1)}(Y - f) + \hat{r}^{(1)}\hat{w}^{(1)}(Y - f) - rw(Y - f)\}^2 \mid \mathcal{D}_2]
$$
$$
\leq 2\mathbb{E}[\{\hat{r}^{(1)}\hat{w}^{(1)}(Y - \hat{f}^{(1)}) - \hat{r}^{(1)}\hat{w}^{(1)}(Y - f)\}^2 \mid \mathcal{D}_2] + 2\mathbb{E}[\{\hat{r}^{(1)}\hat{w}^{(1)}(Y - f) - rw(Y - f)\}^2 \mid \mathcal{D}_2]
$$
$$
\leq 2C_1C_2\|f - \hat{f}^{(1)}\|_2^2 + 2 \times 4R_{\max}^2\|\hat{r}^{(1)}\hat{w}^{(1)} - rw\|_2^2 = \text{o}_p(1).
$$

Here, we have used a parallelogram law from the second line to the third line. We have use $0 < \hat{r} < C_1, 0 < \hat{w} < C_2, |\hat{f}| < R_{\max}$ according to the Assumption 2 and convergence rate conditions, from the third line to the fourth line. The second equation (16) is proved by Jensen's inequality.

**Step 2:**  Eq. (12) is $\text{o}_p(1)$.

We have

$$|\mathbb{E}[\phi_1(X, A, Y; \hat{r}^{(1)}, \hat{w}^{(1)}, \hat{f}^{(1)}) \mid \hat{r}^{(1)}, \hat{w}^{(1)}, \hat{f}] + \mathbb{E}[\phi_2(Z; \hat{f}^{(1)}) \mid \hat{f}] - \mathbb{E}[\phi_1(x; r, w, f)] - \mathbb{E}[\phi_2(Z; f)]|$$
$$\leq |\mathbb{E}[\{\hat{r}^{(1)}(X)\hat{w}^{(1)}(A, X) - r(X)w(A, X)\}\{-\hat{f}^{(1)}(A, X) + f(A, X)\} \mid \hat{r}^{(1)}, \hat{w}^{(1)}, \hat{f}^{(1)}]|$$
$$+ |\mathbb{E}[r(x)w(A, X)\{-\hat{f}(A, X) + f(A, X)\} \mid \hat{r}^{(1)}, \hat{w}^{(1)}, \hat{f}^{(1)}] + \mathbb{E}[\hat{v}^{(1)}(Z) - v(Z) \mid \hat{f}^{(1)}]|$$
$$+ |\mathbb{E}[\hat{r}^{(1)}(X)\hat{w}^{(1)}(A, X)\{Y - f(A, X)\} \mid \hat{r}^{(1)}, \hat{w}^{(1)}]|$$
$$\leq \|\hat{r}^{(1)}(X)\hat{w}^{(1)}(A, X) - r(X)w(A, X)\|_2 \|\hat{f}^{(1)}(A, X) - f(A, X)\|_2 + 0 + 0$$
$$= \alpha\beta + 0 + 0 = \mathrm{o}_p(n^{-1/2}).$$

Here, we have used Hölder's inequality:

$$\|fg\|_1 \leq \|f\|_2 \|g\|_2,$$

the relation

$$\mathbb{E}[r(X)w(A, X)\{-\hat{f}^{(1)}(A, X) + f(A, X)\} \mid \hat{r}^{(1)}, \hat{w}^{(1)}, \hat{f}^{(1)}] + \mathbb{E}[\hat{v}^{(1)}(Z) - v(Z) \mid \hat{f}^{(1)}]$$
$$= \mathbb{E}[-r(X)w(A, X)\hat{f}^{(1)}(A, X) + \hat{v}^{(1)}(z) \mid \hat{r}^{(1)}, \hat{w}^{(1)}, \hat{f}^{(1)}] + \mathbb{E}[-r(X)w(A, X)f(A, X) + v(Z)]$$
$$= 0 + 0 = 0,$$

and

$$\mathbb{E}[\hat{r}^{(1)}(X)\hat{w}^{(1)}(A, X)\{Y - f(A, X)\} \mid \hat{r}^{(1)}, \hat{w}^{(1)}]$$
$$= \mathbb{E}[\hat{r}^{(1)}(X)\hat{w}^{(1)}(A, X)\{f(A, X) - f(A, X)\} \mid \hat{r}^{(1)}, \hat{w}^{(1)}] = 0.$$

**Step 3:**  By combining everything, we have

$$\mathbb{E}_{n_1^{\mathrm{hst}}}[\phi_1(X, A, Y; \hat{r}^{(1)}, \hat{w}^{(1)}, \hat{f}^{(1)})] + \mathbb{E}_{n_1^{\mathrm{evl}}}[\phi_2(Z; \hat{f}^{(1)})] - R(\pi^{\mathrm{e}})$$
$$= \mathbb{E}_{n_1^{\mathrm{hst}}}[\phi_1(X, A, Y; r, w, f)] + \mathbb{E}_{n_1^{\mathrm{evl}}}[\phi_2(Z; f)] - R(\pi^{\mathrm{e}}) + \mathrm{o}_p(1/\sqrt{n}).$$

Then,

$$\hat{R}_{\mathrm{DRCS}} = 0.5\mathbb{E}_{n_1^{\mathrm{hst}}}[\phi_1(X, A, Y; \hat{r}^{(1)}, \hat{w}^{(1)}, \hat{f}^{(1)})] + 0.5\mathbb{E}_{n_1^{\mathrm{evl}}}[\phi_2(Z; \hat{f}^{(1)})]$$
$$+ 0.5\mathbb{E}_{n_2^{\mathrm{hst}}}[\phi_1(X, A, Y; \hat{r}^{(2)}, \hat{w}^{(2)}, \hat{f}^{(2)})] + 0.5\mathbb{E}_{n_2^{\mathrm{evl}}}[\phi_2(Z; \hat{f}^{(2)})]$$
$$= 0.5\mathbb{E}_{n_1^{\mathrm{hst}}}[\phi_1(X, A, Y; r, w, f)] + 0.5\mathbb{E}_{n_1^{\mathrm{evl}}}[\phi_2(Z; f)]+$$
$$+ 0.5\mathbb{E}_{n_2^{\mathrm{hst}}}[\phi_1(X, A, Y; r, w, f)] + 0.5\mathbb{E}_{n_2^{\mathrm{evl}}}[\phi_2(Z; f)] + \mathrm{o}_p(1/\sqrt{n})$$
$$= \mathbb{E}_{n^{\mathrm{hst}}}[\phi_1(X, A, Y; r, w, f)] + \mathbb{E}_{n^{\mathrm{evl}}}[\phi_2(Z; f)] + \mathrm{o}_p(1/\sqrt{n}).$$

Finally, by using a stratified sampling CLT (Wooldridge, 2001), the statement is concluded based on Assumption 1. $\qquad\square$

### E.4   Proof of Lemma 1

*Proof.*  We can bound $\|\hat{r}(X)\hat{w}(A, X) - r(x)w(A, X)\|_2 = \mathrm{o}_p(n^{-p})$:

$$\|\hat{r}(X)\hat{w}(A, X) - r(X)w(A, X)\|_2 \leq \|\hat{r}(X)\hat{w}(A, X) - \hat{r}(X)w(A, X)\|_2$$
$$+ \|\hat{r}(X)w(A, X) - r(X)w(A, X)\|_2$$
$$\leq C_1\mathrm{o}_p(n^{-p}) + C_2\mathrm{o}_p(n^{-p}) = \mathrm{o}_p(n^{-p}).$$

Here, we used the assumptions that $r(X)$ is uniformly bounded by $C_1$ and $w(A, X)$ is uniformly bounded by $C_2$. $\qquad\square$

### E.5   Proof of Theorem 3

*Proof.*  Let us define $\phi_1(x, a, y; r, w, f)$ and $\phi_2(z; f)$:

$$\phi_1(x, a, y; r, w, f) = r(x)w(a, x)\{y - f(a, x)\}, \ \phi_2(z; f) = v(z). \tag{17}$$

We also denote the union of $n_i^{\mathrm{hst}}$ and $n_i^{\mathrm{evl}}$ by $\mathcal{D}_i$ for $i = 1, 2$, and the number of $n_1^{\mathrm{hst}}, n_2^{\mathrm{hst}}, n_1^{\mathrm{evl}}, n_2^{\mathrm{evl}}$ by $n_{11}, n_{21}, n_{12}, n_{22}$. For simplicity, we assume $n_{11} = n_{12}, n_{21} = n_{22}$.

Then, we have

$$\{\mathbb{E}_{n_1^{\mathrm{hst}}}[\phi_1(X, A, Y; \hat{r}^{(1)}, \hat{w}^{(1)}, \hat{f}^{(1)})] + \mathbb{E}_{n_1^{\mathrm{evl}}}[\phi_2(Z; \hat{f}^{(1)})] - R(\pi^{\mathrm{e}})\}$$

$$= \left\{ \frac{1}{\sqrt{n_{11}}} \mathbb{G}_{n_1^{\mathrm{hst}}}[\phi_1(X, A, Y; \hat{r}^{(1)}, \hat{w}^{(1)}, \hat{f}^{(1)}) - \phi_1(X, A, Y; r^\dagger, w^\dagger, f^\dagger)] + \frac{1}{\sqrt{n_{12}}} \mathbb{G}_{n_1^{\mathrm{evl}}}[\phi_2(Z; \hat{f}^{(1)}) - \phi_2(Z; f^\dagger)] \right\} +$$

(18)

$$+ \{\mathbb{E}[\phi_1(X, A, Y; \hat{r}^{(1)}, \hat{w}^{(1)}, \hat{f}^{(1)}) \mid \hat{r}^{(1)}, \hat{w}^{(1)}, \hat{f}^{(1)}] + \mathbb{E}[\phi_2(Z; \hat{f}^{(1)}) \mid \hat{f}^{(1)}] \tag{19}$$

$$- \mathbb{E}[\phi_1(X, A, Y; r^\dagger, w^\dagger, f^\dagger)] - \mathbb{E}[\phi_2(Z; f^\dagger)]\}$$

$$+ \{\mathbb{E}_{n_1^{\mathrm{hst}}}[\phi_1(X, A, Y; r^\dagger, w^\dagger, f^\dagger)] + \mathbb{E}_{n_1^{\mathrm{evl}}}[\phi_2(Z; f^\dagger)] - R(\pi^{\mathrm{e}})\}. \tag{20}$$

The term (18) is $\mathrm{o}_p(1/\sqrt{n})$ by Step 1 in the previous theorem noting that what we have used is $\|\hat{r}(X)\hat{w}(A, X) - w^\dagger(A, X)r^\dagger(X)\| = \mathrm{o}_p(1)$, $\|\hat{f}(A, X) - f^\dagger(A, X)\| = \mathrm{o}_p(1)$. The term Eq. (19) is also $\mathrm{o}_p(1)$ by Step 1 as we will show soon.

**Step 1:**   Eq. (19) is $\mathrm{o}_p(1)$. We have

$$|\mathbb{E}[\phi_1(X, A, Y; \hat{r}^{(1)}, \hat{w}^{(1)}, \hat{f}^{(1)}) \mid \hat{r}^{(1)}, \hat{w}^{(1)}, \hat{f}^{(1)}] + \mathbb{E}[\phi_2(Z; \hat{f}^{(1)}) \mid \hat{f}^{(1)}] - \mathbb{E}[\phi_1(x; r, w, f)] - \mathbb{E}[\phi_2(Z; f)]|$$

$$\leq |\mathbb{E}[\{\hat{r}^{(1)}(X)\hat{w}^{(1)}(A, X) - r^\dagger(X)w^\dagger(A, X)\}\{-\hat{f}^{(1)}(A, X) + f^\dagger(A, X)\} \mid \hat{r}^{(1)}, \hat{w}^{(1)}, \hat{f}^{(1)}]|$$

$$+ |\mathbb{E}[r^\dagger(x)w^\dagger(A, X)\{-\hat{f}^{(1)}(A, X) + f^\dagger(A, X)\} \mid \hat{r}^{(1)}, \hat{w}^{(1)}, \hat{f}^{(1)}] + \mathbb{E}[\hat{v}^{(1)}(Z) - v^\dagger(Z) \mid \hat{f}^{(1)}]|$$

$$+ |\mathbb{E}[\hat{r}^{(1)}(X)\hat{w}^{(1)}(A, X)\{Y - f^\dagger(A, X)\} \mid \hat{r}^{(1)}, \hat{w}^{(1)}].$$

Here, if $f^\dagger(a, x) = f(a, x)$, we have

$$\mathrm{o}_p(1)\mathrm{o}_p(1) + \mathrm{o}_p(1) + 0 = \mathrm{o}_p(1) = \mathrm{o}_p(1).$$

if $r^\dagger(x)w^\dagger(a, x) = r(x)w(a, x)$, we have

$$\mathrm{o}_p(1)\mathrm{o}_p(1) + 0 + \mathrm{o}_p(1) = \mathrm{o}_p(1) = \mathrm{o}_p(1).$$

Therefore, Eq. (19) is $\mathrm{o}_p(1)$.

**Step 2:**   By combining togather, we have

$$\mathbb{E}_{n_1^{\mathrm{hst}}}[\phi_1(X, A, Y; \hat{r}^{(1)}, \hat{w}^{(1)}, \hat{f}^{(1)})] + \mathbb{E}_{n_1^{\mathrm{evl}}}[\phi_2(Z; \hat{f}^{(1)})] - R(\pi^{\mathrm{e}})$$

$$= \mathbb{E}_{n_1^{\mathrm{hst}}}[\phi_1(X, A, Y; r^\dagger, w^\dagger, f^\dagger)] + \mathbb{E}_{n_1^{\mathrm{evl}}}[\phi_2(Z; f^\dagger)] - R(\pi^{\mathrm{e}}) + \mathrm{o}_p(1).$$

Then,

$$\hat{R}_{\mathrm{DRCS}} = 0.5\mathbb{E}_{n_1^{\mathrm{hst}}}[\phi_1(X, A, Y; \hat{r}^{(1)}, \hat{w}^{(1)}, \hat{f}^{(1)})] + 0.5\mathbb{E}_{n_1^{\mathrm{evl}}}[\phi_2(Z; \hat{f}^{(1)})]$$

$$+ 0.5\mathbb{E}_{n_2^{\mathrm{hst}}}[\phi_1(X, A, Y; \hat{r}^{(2)}, \hat{w}^{(2)}, \hat{f}^{(2)})] + 0.5\mathbb{E}_{n_2^{\mathrm{evl}}}[\phi_2(Z; \hat{f}^{(2)})]$$

$$= 0.5\mathbb{E}_{n_1^{\mathrm{hst}}}[\phi_1(X, A, Y; r^\dagger, w^\dagger, f^\dagger)] + 0.5\mathbb{E}_{n_1^{\mathrm{evl}}}[\phi_2(Z; f^\dagger)]+$$

$$+ 0.5\mathbb{E}_{n_2^{\mathrm{hst}}}[\phi_1(X, A, Y; r^\dagger, w^\dagger, f^\dagger)] + 0.5\mathbb{E}_{n_2^{\mathrm{evl}}}[\phi_2(Z; f^\dagger)] + \mathrm{o}_p(1)$$

$$= \mathbb{E}_{n^{\mathrm{hst}}}[\phi_1(X, A, Y; r^\dagger, w^\dagger, f^\dagger)] + \mathbb{E}_{n^{\mathrm{evl}}}[\phi_2(Z; f^\dagger)] + \mathrm{o}_p(1).$$

Then, the statement is concluded because

$$\mathbb{E}\left[\mathbb{E}_{n^{\mathrm{hst}}}[\phi_1(X, A, Y; r^\dagger, w^\dagger, f^\dagger)] + \mathbb{E}_{n^{\mathrm{evl}}}[\phi_2(Z; f^\dagger)]\right] = R(\pi^{\mathrm{e}})$$

based on the double robust structure and

$$\mathbb{E}_{n^{\mathrm{hst}}}[\phi_1(X, A, Y; r^\dagger, w^\dagger, f^\dagger)] + \mathbb{E}_{n^{\mathrm{evl}}}[\phi_2(Z; f^\dagger)] = R(\pi^{\mathrm{e}}) + \mathrm{o}_p(1)$$

from the law of large numbers based on Assumption 1. □

### E.6 Proof of Theorem 14

*Proof.* We can prove similarly as in the proof of Theorem 1. Therefore, we omit the proof.

$\square$

### E.7 Proof of Theorem 4

*Proof of Theorem 4.* For the ease of notation, we prove the case $n^{\text{hst}} = n^{\text{evl}}$ noting the kernel estimator is linearized as in Theorem 13 and and the generalization is easy. We have

$$\left\| \frac{\hat{q}_h(x)}{\hat{p}_h(x)} - \frac{q(x)}{p(x)} - \hat{e}_h(x) \right\|_\infty = \text{o}_p(n^{-1/2}),$$

where

$$\hat{e}_h(x) = \frac{1}{p(x)}\{\hat{q}_h(x) - q(x)\} - \frac{q(x)}{p^2(x)}\{\hat{p}_h(x) - p(x)\}.$$

This is proved by Theorem 12. Then,

$$\hat{R}_{\text{IPWCSB}} = \mathbb{E}_{n^{\text{hst}}}\left[ \frac{\hat{q}_h(X)}{\hat{p}_h(X)} \frac{\pi^{\text{e}}(A \mid X)Y}{\pi^{\text{b}}(A \mid X)} \right]$$

$$= \frac{1}{n^{\text{hst}}} \sum_{i=1}^{n^{\text{hst}}} \frac{\pi^{\text{e}}(A_i \mid X_i)Y_i}{\pi^{\text{b}}(A_i \mid X_i)} \{r(X_i) + \hat{e}_h(X_i)\}$$

$$= \mathbb{E}_{n^{\text{hst}}}\left[ \frac{q(X)}{p(X)} \frac{\pi^{\text{e}}(A \mid X)Y}{\pi^{\text{b}}(A \mid X)} \right] + \frac{1}{n^{\text{hst}}n^{\text{evl}}} \sum_{i=1}^{n^{\text{hst}}} \sum_{j=1}^{n^{\text{evl}}} a_{i,j}$$

$$= \mathbb{E}_{n^{\text{hst}}}\left[ \frac{q(X)}{p(X)} \frac{\pi^{\text{e}}(A \mid X)Y}{\pi^{\text{b}}(A \mid X)} \right] + \frac{2}{n^{\text{hst}}n^{\text{evl}}} \sum_{i<j} b_{i,j},$$

where

$$a_{i,j}((X_i, A_i, Y_i), (Z_j)) = \frac{1}{p(X_i)} \frac{\pi^{\text{e}}(A_i \mid X_i)Y_i}{\pi^{\text{b}}(A_i \mid X_i)}\{K_h(Z_j - X_i) - q(X_i)\}$$

$$- \frac{q(X_i)}{p^2(X_i)} \frac{\pi^{\text{e}}(A_i \mid X_i)Y_i}{\pi^{\text{b}}(A_i \mid X_i)}\{K_h(X_j - X_i) - p(X_i)\},$$

$$b_{i,j}((X_i, A_i, Y_i, Z_i), (X_j, A_j, Y_j, Z_j)) = 0.5\{a_{i,j} + a_{j,i}\}.$$

Then,

$$\frac{2}{n^{\text{hst}}n^{\text{evl}}} \sum_{i<j} b_{i,j}(X_i, A_i, Y_i, Z_i), (X_j, A_j, Y_j, Z_j))$$

$$= \frac{2}{n^{\text{hst}}} \left\{ \sum_{i=1}^{n^{\text{hst}}} \mathbb{E}[b_{i,j} \mid X_i, A_i, Y_i, Z_i] \right\} + \text{o}_p(n^{-1/2})$$

$$= \frac{1}{n^{\text{evl}}} \sum_{i=1}^{n^{\text{evl}}} \mathbb{E}[a_{j,i} \mid X_i, A_i, Y_i, Z_i] + \frac{1}{n^{\text{hst}}} \sum_{i=1}^{n^{\text{hst}}} \mathbb{E}[a_{i,j} \mid X_i, A_i, Y_i, Z_i] + \text{o}_p(n^{-1/2})$$

$$= \frac{1}{n^{\text{evl}}} \sum_{i=1}^{n^{\text{evl}}} \{v(Z_i) - r(X_i)v(X_i)\} + \text{o}_p(n^{-1/2}).$$

From the first line to the second line, we have used a U-statistics theory (van der Vaart, 1998, Chapter 12). From the third line to the fourth line, we have used

$$\mathbb{E}[a_{j,i} \mid Z_i, X_i, A_i, Y_i]$$

$$= \text{o}_p(n^{-1/2}) + \left\{ \mathbb{E}\left[ \frac{1}{p(X_i)} \frac{\pi^{\text{e}}(A_i \mid X_i)Y_i}{\pi^{\text{b}}(A_i \mid X_i)} \mid X_i = Z_i \right] - \mathbb{E}\left[ \frac{q(X_i)}{p^2(X_i)} \frac{\pi^{\text{e}}(A_i \mid X_i)Y_i}{\pi^{\text{b}}(A_i \mid X_i)} \mid X_i \right] \right\} p(X_i)$$

$$= \text{o}_p(n^{-1/2}) + v(Z_i) - r(X_i)v(X_i),$$

$$\mathbb{E}[a_{i,j} \mid Z_i, X_i, A_i, Y_i] = \text{o}_p(n^{-1/2}).$$

Therefore,

$$\hat{R}_{\text{IPWCSB}} = \mathbb{E}_{n^{\text{hst}}} \left[ \frac{q(X)}{p(X)} \left\{ \frac{\pi^{\text{e}}(A \mid X)Y}{\pi^{\text{b}}(A \mid X)} - v(X) \right\} \right] + \mathbb{E}_{n^{\text{evl}}}[v(Z)] + \mathrm{o}_p(n^{-1/2}).$$

The final statement is concluded by CLT. $\qquad\square$

## E.8 Proof of Theorem 5

*Proof.* For the ease of the notation, we prove the case $n^{\text{hst}} = n^{\text{evl}}$ noting the kernel estimator is linearized as in Theorem 13 and the generalization is easy. Here, we concatenate $X$ and $A$ as $D$. We also write $p(x)\pi^{\text{b}}(a \mid x)$ as $u(d)$.

$$\left\| \frac{\hat{q}_h(x)}{\hat{u}_h(d)} - \frac{q(x)}{u(d)} - \hat{e}_h(d) \right\|_\infty = \mathrm{o}_p(n^{-1/2}),$$

where

$$\hat{e}_h(d) = \tfrac{1}{u(d)}\{\hat{q}_h(x) - q(x)\} - \tfrac{q(x)}{u^2(d)}\{\hat{u}_h(d) - u(d)\}.$$

This is proved by Theorem 12. Then,

$$\hat{R}_{\text{IPWCS}} = \mathbb{E}_{n^{\text{hst}}} \left[ \frac{\hat{q}_h(X)}{\hat{p}_h(X)} \frac{\pi^{\text{e}}(A \mid X)Y}{\hat{\pi}_h^b(A \mid X)} \right]$$

$$= \frac{1}{n^{\text{hst}}} \sum_{i=1}^{n^{\text{hst}}} \pi^{\text{e}}(A_i \mid X_i)Y_i \left\{ \frac{q(X_i)}{u(D_i)} + \hat{e}_h(X_i) \right\}$$

$$= \mathbb{E}_{n^{\text{hst}}} \left[ \frac{q(X)}{p(X)} \frac{\pi^{\text{e}}(A \mid X)Y}{\pi^{\text{b}}(A \mid X)} \right] + \frac{1}{n^{\text{hst}}n^{\text{evl}}} \sum_{i=1}^{n^{\text{hst}}} \sum_{j=1}^{n^{\text{evl}}} a_{i,j}$$

$$= \mathbb{E}_{n^{\text{hst}}} \left[ \frac{q(X)}{p(X)} \frac{\pi^{\text{e}}(A \mid X)Y}{\pi^{\text{b}}(A \mid X)} \right] + \frac{2}{n^{\text{hst}}n^{\text{evl}}} \sum_{i<j} b_{i,j},$$

where

$$a_{i,j}((X_i, A_i, Y_i),(Z_j)) = \frac{1}{p(X_i)} \frac{\pi^{\text{e}}(A_i \mid X_i)Y_i}{\pi^{\text{b}}(A_i \mid X_i)} \{K_h(Z_j - X_i) - q(X_i)\}$$

$$- \frac{q(X_i)}{u^2(D_i)} \pi^{\text{e}}(A_i \mid X_i)Y_i\{K_h(D_j - D_i) - u(D_i)\},$$

$$b_{i,j}((X_i, A_i, Y_i, Z_i),(X_j, A_j, Y_j, Z_j)) = 0.5\{a_{i,j} + a_{j,i}\}.$$

Then,

$$\frac{2}{n^{\text{hst}}n^{\text{evl}}} \sum_{i<j} b_{i,j}((X_i, A_i, Y_i, Z_i),(X_j, A_j, Y_j, Z_j))$$

$$= \frac{2}{n^{\text{hst}}} \left\{ \sum_{i=1}^{n^{\text{hst}}} \mathbb{E}[b_{i,j} \mid X_i, A_i, Y_i, Z_i] \right\} + \mathrm{o}_p(n^{-1/2})$$

$$= \frac{1}{n^{\text{evl}}} \sum_{i=1}^{n^{\text{evl}}} \mathbb{E}[a_{j,i} \mid X_i, A_i, Y_i, Z_i] + \frac{1}{n^{\text{hst}}} \sum_{i=1}^{n^{\text{hst}}} \mathbb{E}[a_{i,j} \mid X_i, A_i, Y_i, Z_i] + \mathrm{o}_p(n^{-1/2})$$

$$= \frac{1}{n^{\text{evl}}} \sum_{i=1}^{n^{\text{evl}}} \{v(Z_i) - r(X_i)w(X_i, A_i)f(D_i)\} + \mathrm{o}_p(n^{-1/2}).$$

From the first line to the second line, we used the U-statistics theory (van der Vaart, 1998, Chapter 12). From the third line to the fourth line, we used

$$\mathbb{E}[a_{j,i} \mid Z_i, X_i, A_i, Y_i]$$

$$= \mathrm{o}_p(n^{-1/2}) + \mathbb{E} \left[ \frac{1}{p(X_i)} w(A_i, X_i)Y_i \mid X_i = Z_i \right] p(X_i) - \mathbb{E} \left[ \frac{q(X_i)}{u^2(D_i)} \pi^{\text{e}}(A_i \mid X_i)Y_i \mid D_i = D_i \right] u(D_i)$$

$$= \mathrm{o}_p(n^{-1/2}) + v(Z_i) - r(X_i)w(X_i, A_i)f(D_i),$$

$$\mathbb{E}[a_{i,j} \mid Z_i, X_i, A_i, Y_i] = \mathrm{o}_p(n^{-1/2}).$$

Therefore,

$$\hat{R}_{\text{IPWCS}} = \mathbb{E}_{n^{\text{hst}}}\left[\frac{q(X)}{p(X)}\frac{\pi^{\text{e}}(A \mid X)}{\pi^{\text{b}}(A \mid X)}\{Y - f(A, X)\}\right] + \mathbb{E}_{n^{\text{evl}}}[v(Z)] + o_p(n^{-1/2}).$$

The final statement is concluded by CLT. □

## E.9 Proof of Theorem 6

*Proof.* For the ease of the notation, we prove the case $n^{\text{hst}} = n^{\text{evl}}$ noting the kernel estimator is linearized as in Theorem 13 and and generalization is easy. Here, $\hat{v}_h(a, x)$ is defined as

$$\hat{v}_h(a, x) = \frac{\hat{p}_h(a, x)}{\hat{u}_h(a, x)}, \; \hat{p}_h(a, x) = \frac{1}{n^{\text{hst}}}\sum_{i=1}^{n^{\text{hst}}}Y_iK_h(\{X_i, A_i\} - \{x, a\})$$

$$\hat{u}_h(a, x) = \frac{1}{n^{\text{hst}}}\sum_{i=1}^{n^{\text{hst}}}K_h(\{X_i, A_i\} - \{x, a\}).$$

We have

$$\left\|\frac{\hat{p}_h(a,x)}{\hat{u}_h(a,x)} - \frac{f(a,x)u(a,x)}{u(a,x)} - \hat{e}_h(x, a)\right\|_{\infty} = o_p(n^{-1/2}),$$

where

$$\hat{e}_h(x, a) = \frac{1}{u(a,x)}\{\hat{p}_h(a, x) - f(a, x)u(a, x)\} - \frac{f(a,x)}{u(a,x)}\{\hat{u}_h(a, x) - u(a, x)\}.$$

This is proved by Theorem 12. Then, we have

$$\hat{R}_{\text{DM}} = \mathbb{E}_{n^{\text{evl}}}[\hat{v}_h(Z, A)]$$

$$= \mathbb{E}_{n^{\text{evl}}}[v(Z, A)] + \frac{1}{n^{\text{hst}}n^{\text{evl}}}\sum_{i=1}^{n^{\text{evl}}}\sum_{j=1}^{n^{\text{hst}}}a_{i,j}$$

$$= \mathbb{E}_{n^{\text{evl}}}[v(Z, A)] + \frac{2}{n^{\text{hst}}n^{\text{evl}}}\sum_{i<j}b_{i,j},$$

where

$$a_{i,j}((Z_i, A_i), (X_j, A_j, Y_j)) = \frac{1}{u(Z_i, A_i)}\{Y_jK_h(\{X_j, A_j\} - \{Z_i, A_i\}) - u(Z_i, A_i)f(Z_i, A_i)\} -$$

$$\frac{f(Z_i, A_i)}{u(Z_i, A_i)}\{K_h(\{X_j, A_j\} - \{Z_i, A_i\}) - u(Z_i, A_i)\},$$

$$b_{i,j}((X_i, A_i, Y_i, Z_i), (X_j, A_j, Y_j, Z_j)) = 0.5\{a_{i,j} + a_{j,i}\}.$$

Then,

$$\frac{2}{n^{\text{hst}}n^{\text{evl}}}\sum_{i<j}b_{i,j}(X_i, A_i, Y_i, Z_i), (X_j, A_j, Y_j, Z_j))$$

$$= \frac{2}{n^{\text{hst}}}\left\{\sum_{i=1}^{n^{\text{hst}}}\mathbb{E}[b_{i,j} \mid X_i, A_i, Y_i, Z_i]\right\} + o_p(n^{-1/2})$$

$$= \frac{1}{n^{\text{evl}}}\sum_{i=1}^{n^{\text{evl}}}\mathbb{E}[a_{j,i} \mid X_i, A_i, Y_i, Z_i] + \frac{1}{n^{\text{hst}}}\sum_{i=1}^{n^{\text{hst}}}\mathbb{E}[a_{i,j} \mid X_i, A_i, Y_i, Z_i] + o_p(n^{-1/2})$$

$$= \frac{1}{n^{\text{evl}}}\sum_{i=1}^{n^{\text{evl}}}\{Y_i - f(X_i, A_i)\}r(X_i)w(X_i, A_i) + o_p(n^{-1/2}).$$

From the first line to the second line, we used a U-statistics theory (van der Vaart, 1998, Chapter 12). From the third line to the fourth line, we used

$$\mathbb{E}[a_{j,i} \mid Z_i, X_i, A_i, Y_i]$$

$$= \mathrm{o}_p(n^{-1/2}) + \left\{ \frac{Y_i}{u(X_i, A_i)} - \frac{f(X_i, A_i)}{u(X_i, A_i)} \right\} q(X_i)\pi^{\mathrm{e}}(A_i \mid X_i)$$

$$= \mathrm{o}_p(n^{-1/2}) + Y_i r(X_i) w(A_i, X_i) - r(X_i) f(X_i, A_i) w(A_i, X_i),$$

$$\mathbb{E}[a_{i,j} \mid Z_i, X_i, A_i, Y_i] = \mathrm{o}_p(n^{-1/2}).$$

This is proved by Theorem 12. Therefore,

$$\hat{R}_{\mathrm{DM}} = \mathbb{E}_{n^{\mathrm{hst}}} \left[ r(X)w(A, X)\{Y - f(A, X)\} \right] + \mathbb{E}_{n^{\mathrm{evl}}}[v(Z)] + \mathrm{o}_p(n^{-1/2}).$$

The final statement is concluded by CLT. $\qquad\square$

### E.10 Proof of Theorem 7

*Proof.* We prove the statement following Zhou et al. (2018). Though the proof is very similar, for completeness, we sketch the proof the case $\rho = 0.5$. Because the estimator is asymptotically linear, the generalization is easy as in Theorem 13.

Define two scores;

$$\hat{\Gamma}_i = \hat{r}_w^{(D_i)}(X_i, A_i)\{Y_i - f^{(D_i)}(X_i, A_i)\} + [f(a_1, X_i), \cdots, f(a_\alpha, X_i)]^\top,$$

$$\Gamma_i = r_w(X_i, A_i)\{Y_i - f(X_i, A_i)\} + [f(a_1, X_i), \cdots, f(a_\alpha, X_i)]^\top,$$

where $D_i$ is an indicator which cross-fold estimator is used and $\alpha$ is a dimension of the action, $r_w(x) = r(x)/\pi^{\mathrm{b}}(a \mid x)$. Then, we have $\hat{R}_{\mathrm{DRCS}}(\pi) = \frac{2}{n}\{\sum_{i=1}^{n/2}\langle \pi(Z_i), \hat{\Gamma}_i \rangle\}$. Here, we define the estimator with oracle efficient influence function

$$\tilde{R}(\pi) = \frac{2}{n}\sum_{i=1}^{n/2}\langle \pi(Z_i), \Gamma_i \rangle.$$

In addition, we define

$$\Delta(\pi_a, \pi_b) = R(\pi_a) - R(\pi_b), \tilde{\Delta}(\pi_a, \pi_b) = \tilde{R}(\pi_a) - \tilde{R}(\pi_b),$$

$$\hat{\Delta}(\pi_a, \pi_b) = \hat{R}(\pi_a) - \hat{R}(\pi_b).$$

**Step 1:** First, following Zhou et al. (2018, Theorem 2), we prove the following. Let $\tilde{\pi} \in \arg\min_{\pi \in \Pi} \tilde{R}(\pi)$. Then for any $\delta > 0$, with probability at least $1 - 2\delta$,

$$R(\tilde{\pi}) - R(\pi^*) \leq O\left( \left\{ k(\Pi) + \sqrt{\log(1/\delta)} \right\} \sqrt{\frac{\Upsilon_*}{n}} \right),$$

where

$$\Upsilon_* = \sup_{\pi \in \Pi} \mathbb{E}[\langle \Gamma_i, \pi(Z_i) \rangle^2]$$

$$= \sup_{\pi \in \Pi} \mathbb{E}[r(X_i)^2 w_\pi^2(X_i, A_i)\{Y_i - f(X_i, A_i)\}^2] + \mathbb{E}[\{v_\pi(Z_i)\}^2],$$

when $w_\pi(a, x) = \pi(a, x)/\pi^{\mathrm{b}}(a, x)$, $v_\pi(x) = \mathbb{E}_{\pi(a|x)}[f(a, x) \mid x]$.

This is proved as follows. We have

$$R(\tilde{\pi}) - R(\pi^*) \leq \sup_{\pi_a, \pi_b \in \Pi} |\tilde{\Delta}(\pi_a, \pi_b) - \Delta(\pi_a, \pi_b)|.$$

Then, by using a Chaining argument as Lemma 1 (Zhou et al., 2018), we can bound an expectation of $\sup_{\pi_a, \pi_b \in \Pi} |\tilde{\Delta}(\pi_a, \pi_b) - \Delta(\pi_a, \pi_b)|$ via Rademacher complexity. Then, as in Lemma 2 (Zhou et al., 2018), the high probability bound is obtained via Talagrand inequality. Then, we have

$$R(\tilde{\pi}) - R(\pi^*) \leq O\left( \left\{ k(\Pi) + \sqrt{\log(1/\delta)} \right\} \sqrt{\frac{\Upsilon_*'}{n}} \right),$$

where

$$\Upsilon'_* = \sup_{\pi_a, \pi_b \in \Pi} \mathbb{E}[\langle \Gamma_i, \pi_a(Z_i) - \pi_b(Z_i) \rangle^2].$$

This concludes the above statement because

$$\Upsilon_* = \sup_{\pi_a, \pi_b \in \Pi} \mathbb{E}[\langle \Gamma_i, \pi_a(Z_i) - \pi_b(Z_i) \rangle^2]$$

$$= \sup_{\pi_a, \pi_b \in \Pi} \mathbb{E}[r(X_i)^2 \{w_{\pi_a} - w_{\pi_b}\}^2 \{Y_i - f(X_i, A_i)\}^2] + \mathbb{E}[\{v_{\pi_a}(Z_i) - v_{\pi_b}(Z_i)\}^2]$$

$$\leq \sup_{\pi \in \Pi} 2\mathbb{E}[r(X_i)^2 w_\pi^2(X_i, A_i)\{Y_i - f(X_i, A_i)\}^2] + 2\mathbb{E}[\{v_\pi(Z_i)\}^2].$$

**Step 2:** Assume $\kappa(\Pi) < \infty$, then

$$\sup_{\pi_a, \pi_b \in \Pi} |\tilde{\Delta}(\pi_a, \pi_b) - \hat{\Delta}(\pi_a, \pi_b)| = o_p(n^{-1/2}).$$

The proof of this statement is based on the double structure of the influence function and cross-fitting. We omit the proof because it is long, and almost the same as Lemma 3 (Zhou et al., 2018).

**Step 3:** Finally, based on Theorem 3 (Zhou et al., 2018), we have

$$R(\hat{\pi}) - R(\pi^*) \leq \sup_{\pi_a, \pi_b \in \Pi} |\tilde{\Delta}(\pi_a, \pi_b) - \hat{\Delta}(\pi_a, \pi_b)| + \sup_{\pi_a, \pi_b \in \Pi} |\tilde{\Delta}(\pi_a, \pi_b) - \Delta(\pi_a, \pi_b)|$$

$$\leq O_p \left( \left\{ k(\Pi) + \sqrt{\log(1/\delta)} \right\} \sqrt{\frac{\Upsilon_*}{n}} \right).$$

This means there exists an integer $N_\delta$ such that with probability at least $1 - 2\delta$, for all $n \geq N_\delta$:

$$R(\hat{\pi}) - R(\pi^*) \lesssim \left( k(\Pi) + \sqrt{\log(1/\delta)} \right) \sqrt{\frac{\Upsilon_*}{n}}.$$

**Remark 8.** In the general case,

$$\Upsilon_* = \sup_{\pi \in \Pi} \rho^{-1} \mathbb{E}[r(X_i)^2 w_\pi^2(X_i, A_i)\{Y_i - f(X_i, A_i)\}^2] + (1 - \rho)^{-1} \mathbb{E}[\{v_\pi(Z_i)\}^2].$$

$\square$

# F  OPE with Known Distribution of Evaluation Data

In this section, we consider a special case where $q(x)$ is known.

By applying (3) in Section 2.3, we obtain the efficiency bound under nonparametric model defined, which is defined as $\tilde{\Upsilon}(\pi^e) = \mathbb{E}[r^2(X)w^2(A, X)\text{var}[Y \mid A, X]]$.

As the estimator $\hat{R}_{\text{DRCS}}(\pi^e)$ in Section 4, we construct an estimator with cross-fitting. Instead of (6), we use an estimator defined as $\mathbb{E}_{n_k^{\text{hst}}}[\hat{r}^{(k)}(X)\hat{w}^{(k)}(A, X)\{Y - \hat{f}^{(k)}(A, X)\}] + \mathbb{E}_{q(z)\pi^e(a|z)}[\hat{f}^{(k)}(a, z)]$. The algorithm is almost the same as before. To estimate $r(x)$, we can simply use density estimation for $p(x)$ because $q(x)$ is known and the integration in $\mathbb{E}_{q(z)\pi^e(a|z)}[\hat{f}(a, z)]$ can be taken exactly because $q(x)$ and $\pi^e(a \mid x)$ are known. Let us denote this estimator as $\tilde{R}_{\text{DRCS}}$. We can show that $\tilde{R}_{\text{DRCS}}(\pi^e)$ achieves the efficiency bound.

**Theorem 14** (Efficiency of $\tilde{R}_{\text{DRCS}}$). *For $k \in \{1, \cdots, \xi\}$, assume there exists $p > 0$, $q > 0$, $p+q \geq 1/2$ such that $\|\hat{r}^{(k)}(X)\hat{w}^{(k)}(A, X) - r(X)w(A, X)\|_2 = o_p(n^{-p})$ and $\|\hat{f}^{(k)}(A, X) - f(A, X)\|_2 = o_p(n^{-q})$. Then, we have $\sqrt{n^{\text{hst}}}(\tilde{R}_{\text{DRCS}}(\pi^e) - R(\pi^e)) \xrightarrow{d} \mathcal{N}(0, \tilde{\Upsilon}(\pi^e))$.*

This asymptotic variance is equal to the asymptotic variance when $\rho = 0$ as shown in Remark 2 because the case $\rho = 0$ implies that we have infinite data from $q(x)$.

Table 5: Specification of datasets

| Dataset | the number of samples | Dimension | the number of classes |
|---------|----------------------|-----------|-----------------------|
| satimage | 4,435 | 35 | 6 |
| vehicle | 846 | 18 | 4 |
| pendigits | 7,496 | 16 | 10 |

## G Algorithm for Off-Policy Learning with Cross-Fitting

In the proposed method of OPL under a covariate shift, we train an evaluation policy by using an estimator $\hat{R}_{\mathrm{DRCS}}(\pi^{\mathrm{e}})$, which is constructed via cross-fitting. In this section, we introduce an algorithm where we use a linear-in-parameter model with kernel functions to approximate a new policy. For $x \in \mathcal{X}$, a linear-in-parameter model is defined as $\pi(a \mid x; \sigma^2) = \frac{\exp(g(a,x;\sigma^2))}{\sum_{a \in \mathcal{A}} \exp(g(a,x;\sigma^2))}$, where $g(a, x; \sigma^2) = \beta_a^\top \varphi(x; \sigma^2) + \beta_{0,a}$, $\varphi(x; \sigma^2) = \left[\varphi_1(x; \sigma^2), \ldots, \varphi_m(x; \sigma^2)\right]^\top$, $\varphi_m(x; \sigma^2)$ is the Gaussian kernel defined as $\varphi_u(x; \sigma^2) = \exp\left(-\frac{\|x - c_u\|^2}{2\sigma^2}\right)$, $1 \le u \le m$, where $\{c_1, ..., c_m\}$ is $m$ chosen points from $\{X_i\}_{i=1}^{n^{\mathrm{hst}}}$, $\beta_a \in \mathbb{R}^m$, and $\beta_{0,a} \in \mathbb{R}$. In optimization, we put a regularization term $\mathcal{R}(\{\beta_a, \beta_{0,a}\})$ and train a new policy as $\hat{\pi}_{\mathrm{DRCS}} = \arg\max_{\pi \in \Pi} \hat{R}_{\mathrm{DRCS}}(\pi) + \lambda \mathcal{R}(\{\beta_a, \beta_{0,a}\})$, where $\lambda > 0$. The parameters $\sigma^2$ and $\lambda$ are hyper-parameters selected via cross-validation. Thus, in the proposed method, we use the cross-fitting and cross-validation. We describe the algorithm in Algorithm 2 with $K$ fold cross-fitting and $L$ fold cross-validation. For brevity, in the algorithm, let us assume $n^{\mathrm{hst}}/\xi, n^{\mathrm{hst}}/L, n^{\mathrm{evl}}/\xi, n^{\mathrm{evl}}/L \in \mathbb{N}$. In Algorithm 2, we express the objective function with hyper-parameters $\sigma^2$ and $\lambda$ as $\mathbb{E}_{n^{\mathrm{hst}}}\left[\hat{r}(X)\frac{\pi(A|X;\sigma^2)}{\hat{\pi}^{\mathrm{b}}(A|X)}\{Y - \hat{f}(A, X)\}\right] + \mathbb{E}_{n^{\mathrm{evl}}}[\mathbb{E}_{\pi^{\mathrm{e}}(a|Z)}[\hat{f}(a, Z)\pi(a \mid Z; \sigma^2)] + \lambda \mathcal{R}(\{\beta_a, \beta_{0,a}\})$.

## H Details of Experiments in Section 7.1

First, we show the description of the datasets in Table. All datasets are downloaded from `https://www.csie.ntu.edu.tw/~cjlin/libsvmtools/datasets/`.

In addition to the results shown in Section 7.1, we show the performances of IPWCS and DM estimator with nuisance functions estimated by the kernel Ridge regression, which are referred as IPWCS-R and DM-R. In addition, for OPE, we also show the results with the different sample size.

In Tables 6–7, we show the additional experimental results with the same setting as Section 7. In this setting, the sample size is fixed at 800.

In Tables 8–9, we show the additional experimental results with 500 samples. The other setting is the same as Section 7.1.

In Tables 10–11, we show the additional experimental results with 300 samples. The other setting is the same as Section 7.1.

We also add the OPE and OPL experiment with the `pendigits` dataset in Tables 12 and 13. In this experiment, the sample size is fixed at 800.

In Tables 14–15, we show the additional experimental results with $1,000$ samples for the `satimage` and `pendigits` datasets. We could not conduct experiments for the `vehicle` dataset because it only has 800 samples. The other setting is the same as Section 7.1.

For OPE, we highlight in bold the best two estimators in each case. For OPL, we highlight in bold the best one estimator in each case. The proposed DRCS estimator performs well in many datasets. The DM estimator also works well, but the performance dramatically drops when the model is misspecified.

**Remark 9** (Self-normalization). We can add self-normalization for improving the performances of the proposed estimator Swaminathan & Joachims (2015a). There can be several ways for incorporating the density ratio estimator in the self-normalization. Because the proposition of self-normalization method is not main topic of this paper, we omit the details.

**Algorithm 2** Off-policy learning using $\hat{R}_{\mathrm{DRCS}}(\pi^{\mathrm{e}})$ with $\xi$-fold cross-fitting.

---

**Input**: $\xi$: the number of the cross-fitting for constructing $\hat{R}_{\mathrm{DRCS}}(\pi^{\mathrm{e}})$. $L$: the number of the cross-validation for constructing the optimal policy. $\Pi$: a hypothesis class of $\pi^{\mathrm{e}}$. $\{\sigma_1^2, \ldots, \sigma_{n_{\sigma^2}}^2\}$: candidates of $\sigma^2$. $\{\lambda_1, \ldots, \lambda_{n_\lambda}\}$: candidates of $\lambda$.

Take a $\xi$-fold random partition $(I_k)_{k=1}^{\xi}$ of observation indices $[n^{\mathrm{hst}}] = \{1, \ldots, n^{\mathrm{hst}}\}$ such that the size of each fold $I_k$ is $n_k^{\mathrm{hst}} = n^{\mathrm{hst}}/\xi$.

Take a $\xi$-fold random partition $(J_k)_{k=1}^{\xi}$ of observation indices $[n^{\mathrm{evl}}] = \{1, \ldots, n^{\mathrm{evl}}\}$ such that the size of each fold $J_k$ is $n_k^{\mathrm{evl}} = n^{\mathrm{evl}}/\xi$.

For each $k \in [\xi] = \{1, \ldots, \xi\}$, define $I_k^c := \{1, \ldots, n^{\mathrm{hst}}\} \setminus I_k$ and $J_k^c := \{1, \ldots, n^{\mathrm{evl}}\} \setminus J_k$.

Define $\mathcal{S}_k = \{(X_i, A_i, Y_i)\}_{i \in I_k^c}$.

**for** $k \in [K]$ **do**

    Construct nuisance estimators $\hat{\pi}_k^{\mathrm{b}}(a \mid X)$, $\hat{r}_k(x)$, and $\hat{f}_k(a, x)$ using $\mathcal{S}_k$.

**end for**

Take a $L$-fold random partition $(I_\ell)_{\ell=1}^{L}$ of observation indices $[n^{\mathrm{hst}}] = \{1, \ldots, n^{\mathrm{hst}}\}$ such that the size of each fold $I_\ell$ is $n_\ell^{\mathrm{hst}} = n^{\mathrm{hst}}/L$.

Take a $L$-fold random partition $(J_\ell)_{\ell=1}^{L}$ of observation indices $[n^{\mathrm{evl}}] = \{1, \ldots, n^{\mathrm{evl}}\}$ such that the size of each fold $J_\ell$ is $n_\ell^{\mathrm{evl}} = n^{\mathrm{evl}}/L$.

For each $\ell \in [L] = \{1, \ldots, L\}$, define $I_\ell^c := \{1, \ldots, n^{\mathrm{hst}}\} \setminus I_\ell$ and $J_\ell^c := \{1, \ldots, n^{\mathrm{evl}}\} \setminus J_\ell$.

**for** $\tilde{\sigma}^2 \in \{\sigma_1^2, \ldots, \sigma_{n_{\sigma^2}}^2\}$ **do**

    **for** $\tilde{\lambda} \in \{\lambda_1, \ldots, \lambda_{n_\lambda}\}$ **do**

        Define $Score_{\tilde{\sigma}^2, \tilde{\lambda}} = 0$.

        **for** $\ell \in [L]$ **do**

            Obtain $\tilde{\pi}$ by solving the following optimization problem:

$$\tilde{\pi} = \arg\max_{\pi \in \Pi} \mathbb{E}_{n_{I_\ell}^{\mathrm{hst}}} \left[ \hat{r}(X) \frac{\pi(A \mid X; \tilde{\sigma}^2)}{\hat{\pi}^{\mathrm{b}}(A \mid X)} \{Y - \hat{f}(A, X)\} \right]$$
$$+ \mathbb{E}_{n_{J_\ell}^{\mathrm{evl}}} [\mathbb{E}_{\pi^{\mathrm{e}}(a \mid Z)} [\hat{f}(a, Z) \pi(a \mid Z; \tilde{\sigma}^2)]] + \tilde{\lambda} \mathcal{R}(\{\beta_a, \beta_{0,a}\}),$$

            where $\mathbb{E}_{n_{I_\ell}^{\mathrm{hst}}}$ denotes a empirical approximation using $i \in I_\ell$, $\mathbb{E}_{n_{J_\ell}^{\mathrm{evl}}}$ denotes a sample approximation using $j \in J_\ell$, and $\hat{\pi}^{\mathrm{b}}$, $\hat{r}$, and $\hat{f}$ are the corresponding nuisance estimators chosen from $\hat{\pi}_k^{\mathrm{b}}$, $\hat{r}_k$, and $\hat{f}_k$.

            Update the score $Score_{\tilde{\sigma}^2, \tilde{\lambda}}$ by

$$Score_{\tilde{\sigma}^2, \tilde{\lambda}}$$
$$= Score_{\tilde{\sigma}^2, \tilde{\lambda}} + \mathbb{E}_{n_{I_\ell^c}^{\mathrm{hst}}} \left[ \hat{r}(X) \frac{\tilde{\pi}(A \mid X; \tilde{\sigma}^2)}{\hat{\pi}^{\mathrm{b}}(A \mid X)} \{Y - \hat{f}(A, X)\} \right] + \mathbb{E}_{n_{J_\ell^c}^{\mathrm{evl}}} [\mathbb{E}_{\pi^{\mathrm{e}}(a \mid Z)} [\hat{f}(a, Z) \tilde{\pi}(a \mid Z; \tilde{\sigma}^2)]],$$

            where $\mathbb{E}_{n_{I_\ell^c}^{\mathrm{hst}}}$ denotes a empirical approximation using $i \in I_\ell^c$, and $\mathbb{E}_{n_{J_\ell^c}^{\mathrm{evl}}}$ denotes a sample approximation using $j \in J_\ell^c$.

        **end for**

    **end for**

**end for**

Obtain $\tilde{\pi}$ by solving the following optimization problem:

$$\hat{\pi} = \arg\max_{\pi \in \Pi} \mathbb{E}_{n^{\mathrm{hst}}} \left[ \hat{r}(X) \frac{\pi(A \mid X; \hat{\sigma}^2)}{\hat{\pi}^{\mathrm{b}}(A \mid X)} \{Y - \hat{f}(A, X)\} \right]$$
$$+ \mathbb{E}_{n_{J_\ell}^{\mathrm{evl}}} [\mathbb{E}_{\pi^{\mathrm{e}}(a \mid Z)} [\hat{f}(a, Z) \pi(a \mid Z; \hat{\sigma}^2)]] + \hat{\lambda} \mathcal{R}(\{\beta_a, \beta_{0,a}\}),$$

where $(\hat{\sigma}^2, \hat{\lambda}) = \arg\max_{(\tilde{\sigma}^2, \tilde{\lambda}) \in \{\{\sigma_1^2, \ldots, \sigma_{n_{\sigma^2}}^2\}, \{\lambda_1, \ldots, \lambda_{n_\lambda}\}\}} Score_{\tilde{\sigma}^2, \tilde{\lambda}}$.

---

Table 6: Off-policy evaluation with the `satimage` dataset with 800 samples

| Behavior Policy | DRCS | | IPWCS | | DM | | IPWCS-R | | DM-R | |
|---|---|---|---|---|---|---|---|---|---|---|
| | MSE | std | MSE | std | MSE | std | MSE | std | MSE | std |
| $0.7\pi^d + 0.3\pi^u$ | 0.107 | 0.032 | 67.448 | 144.845 | **0.042** | 0.043 | **0.045** | 0.049 | 0.073 | 0.023 |
| $0.4\pi^d + 0.6\pi^u$ | **0.096** | 0.025 | 74.740 | 155.704 | 0.134 | 0.052 | **0.093** | 0.069 | 0.177 | 0.033 |
| $0.0\pi^d + 1.0\pi^u$ | **0.154** | 0.051 | 58.031 | 103.632 | 0.336 | 0.079 | **0.022** | 0.026 | 0.372 | 0.050 |

Table 7: Off-policy evaluation with the `vehicle` dataset with 800 samples

| Behavior Policy | DRCS | | IPWCS | | DM | | IPWCS-R | | DM-R | |
|---|---|---|---|---|---|---|---|---|---|---|
| | MSE | std | MSE | std | MSE | std | MSE | std | MSE | std |
| $0.7\pi^d + 0.3\pi^u$ | **0.029** | 0.019 | 218390.000 | 285382.247 | **0.038** | 0.035 | 0.568 | 0.319 | 0.040 | 0.014 |
| $0.4\pi^d + 0.6\pi^u$ | **0.019** | 0.024 | 329825.704 | 454301.175 | 0.095 | 0.062 | 0.576 | 0.357 | **0.089** | 0.019 |
| $0.0\pi^d + 1.0\pi^u$ | **0.037** | 0.030 | 173603.802 | 141163.618 | 0.213 | 0.049 | 0.233 | 0.193 | **0.210** | 0.031 |

Table 8: Off-policy evaluation with the `satimage` dataset with 500 samples

| Behavior Policy | DRCS | | IPWCS | | DM | | IPWCS-R | | DM-R | |
|---|---|---|---|---|---|---|---|---|---|---|
| | MSE | std | MSE | std | MSE | std | MSE | std | MSE | std |
| $0.7\pi^d + 0.3\pi^u$ | 0.112 | 0.039 | 729.208 | 2433.557 | **0.049** | 0.042 | 0.177 | 0.407 | **0.079** | 0.033 |
| $0.4\pi^d + 0.6\pi^u$ | **0.087** | 0.036 | 790.188 | 2139.882 | 0.146 | 0.074 | **0.130** | 0.170 | 0.173 | 0.045 |
| $0.0\pi^d + 1.0\pi^u$ | **0.179** | 0.066 | 453.553 | 1148.372 | 0.335 | 0.097 | **0.047** | 0.071 | 0.374 | 0.070 |

Table 9: Off-policy evaluation with the `vehicle` dataset with 500 samples

| Behavior Policy | DRCS | | IPWCS | | DM | | IPWCS-R | | DM-R | |
|---|---|---|---|---|---|---|---|---|---|---|
| | MSE | std | MSE | std | MSE | std | MSE | std | MSE | std |
| $0.7\pi^d + 0.3\pi^u$ | **0.029** | 0.020 | 104311.242 | 126027.165 | **0.028** | 0.027 | 0.379 | 0.317 | 0.036 | 0.022 |
| $0.4\pi^d + 0.6\pi^u$ | **0.014** | 0.010 | 186170.520 | 260715.112 | **0.081** | 0.040 | 0.585 | 0.553 | 0.082 | 0.031 |
| $0.0\pi^d + 1.0\pi^u$ | **0.034** | 0.038 | 82883.403 | 115580.232 | **0.149** | 0.064 | 0.230 | 0.235 | 0.184 | 0.042 |

Table 10: Off-policy evaluation with the `satimage` dataset with 300 samples

| Behavior Policy | DRCS | | IPWCS | | DM | | IPWCS-R | | DM-R | |
|---|---|---|---|---|---|---|---|---|---|---|
| | MSE | std | MSE | std | MSE | std | MSE | std | MSE | std |
| $0.7\pi^d + 0.3\pi^u$ | 0.103 | 0.043 | 765.985 | 2922.342 | **0.026** | 0.027 | 0.125 | 0.115 | **0.067** | 0.035 |
| $0.4\pi^d + 0.6\pi^u$ | **0.074** | 0.051 | 40.273 | 89.381 | **0.126** | 0.098 | 0.261 | 0.309 | 0.155 | 0.055 |
| $0.0\pi^d + 1.0\pi^u$ | **0.169** | 0.095 | 4367.009 | 15791.530 | **0.297** | 0.084 | 0.375 | 1.293 | 0.341 | 0.073 |

Table 11: Off-policy evaluation with the `vehicle` dataset with 300 samples

| Behavior Policy | DRCS | | IPWCS | | DM | | IPWCS-R | | DM-R | |
|---|---|---|---|---|---|---|---|---|---|---|
| | MSE | std | MSE | std | MSE | std | MSE | std | MSE | std |
| $0.7\pi^d + 0.3\pi^u$ | **0.036** | 0.023 | 78064.888 | 80378.226 | **0.029** | 0.028 | 0.328 | 0.391 | 0.038 | 0.021 |
| $0.4\pi^d + 0.6\pi^u$ | **0.020** | 0.020 | 108655.809 | 136013.160 | 0.096 | 0.055 | 0.668 | 0.608 | **0.084** | 0.033 |
| $0.0\pi^d + 1.0\pi^u$ | **0.063** | 0.051 | 59301.622 | 74435.924 | 0.175 | 0.074 | **0.125** | 0.161 | 0.204 | 0.053 |

Table 12: Off-policy evaluation with `pendigits` dataset with 800 samples

| Behavior Policy | DRCS | | IPWCS | | DM | | IPWCS-R | | DM-R | |
|---|---|---|---|---|---|---|---|---|---|---|
| | MSE | std | MSE | std | MSE | std | MSE | std | MSE | std |
| $0.7\pi^d + 0.3\pi^u$ | 0.118 | 0.020 | 1074.278 | 838.074 | **0.083** | 0.035 | **0.052** | 0.045 | 0.089 | 0.014 |
| $0.4\pi^d + 0.6\pi^u$ | **0.110** | 0.026 | 1328.069 | 1045.287 | 0.220 | 0.053 | **0.056** | 0.040 | 0.231 | 0.026 |
| $0.0\pi^d + 1.0\pi^u$ | **0.314** | 0.086 | 231.043 | 217.068 | 0.503 | 0.049 | **0.116** | 0.187 | 0.511 | 0.037 |

Table 13: Off-policy learning with `pendigits` dataset with 800 samples

| Behavior Policy | DRCS | | IPWCS | | DM | |
|---|---|---|---|---|---|---|
| | RWD | STD | RWD | STD | RWD | STD |
| $0.7\pi^d + 0.3\pi^u$ | **0.683** | 0.030 | 0.241 | 0.048 | 0.507 | 0.060 |
| $0.4\pi^d + 0.6\pi^u$ | **0.678** | 0.039 | 0.252 | 0.032 | 0.445 | 0.096 |
| $0.0\pi^d + 1.0\pi^u$ | **0.409** | 0.067 | 0.204 | 0.031 | 0.212 | 0.041 |

Table 14: Off-policy evaluation with the `satimage` dataset with $1,000$ samples

| Behavior Policy | DRCS | | IPWCS | | DM | | IPWCS-R | | DM-R | |
|---|---|---|---|---|---|---|---|---|---|---|
| | MSE | std | MSE | std | MSE | std | MSE | std | MSE | std |
| $0.7\pi^d + 0.3\pi^u$ | 0.111 | 0.024 | 58.724 | 91.964 | **0.052** | 0.034 | **0.050** | 0.069 | 0.067 | 0.019 |
| $0.4\pi^d + 0.6\pi^u$ | **0.090** | 0.026 | 118.317 | 188.729 | 0.173 | 0.097 | **0.099** | 0.087 | 0.170 | 0.039 |
| $0.0\pi^d + 1.0\pi^u$ | **0.145** | 0.038 | 82.801 | 103.326 | 0.369 | 0.106 | **0.018** | 0.026 | 0.395 | 0.046 |

Table 15: Off-policy evaluation with the `pendigits` dataset with $1,000$ samples

| Behavior Policy | DRCS | | IPWCS | | DM | | IPWCS-R | | DM-R | |
|---|---|---|---|---|---|---|---|---|---|---|
| | MSE | std | MSE | std | MSE | std | MSE | std | MSE | std |
| $0.7\pi^d + 0.3\pi^u$ | 0.118 | 0.021 | 1299.936 | 829.752 | 0.094 | 0.029 | **0.040** | 0.040 | **0.090** | 0.012 |
| $0.4\pi^d + 0.6\pi^u$ | **0.106** | 0.021 | 1483.730 | 1014.923 | 0.256 | 0.063 | **0.067** | 0.078 | 0.241 | 0.023 |
| $0.0\pi^d + 1.0\pi^u$ | **0.313** | 0.091 | 300.599 | 216.541 | 0.496 | 0.064 | **0.099** | 0.167 | 0.531 | 0.033 |

## Footnotes

[7]The reason why we use $B$ is to distinguish it from the random variable $A$.