[Reviews · NeurIPS 2020]

Review 1

Summary and Contributions: The authors propose a doubly robust method for Off-Policy Evaluation and Off-Policy Learning in bandit (point-treatment) problems where (1) the evaluation policy differs from the behavior policy (observational setting), and (2) the distribution of contexts in in the logged historical data differs from the distribution of contexts (when the target population changes). The method utilizes reweighting to correct for covariate shift; reweighting to correct for policy shift; and a direct model of the reward function. The authors work out the semiparametric efficiency lower bound in this context, and show that their doubly robust estimator achieves this lower bound under weaker assumptions than methods that use the two reweighting adjustments alone, or methods that use direct estimation alone. They also derive a regret bound for off-policy learning in this context, and show that their estimator achieves it. Finally, the authors demonstrate that their method is more effective than simpler approaches in a finite-sample setting.

Strengths: The authors thoroughly map out the semiparametric efficiency theory in this setting, and show that an extension of standard doubly robust estimators is effective here. The thoroughness of results and clarity of exposition is impressive.

Weaknesses: In general, it would be nice to have a bit more insight into the theoretical approach in the main text. In particular, it would be nice to better understand exactly what changes in the theoretical approach between the covariate shift and no-shift settings. I understand that many of the results in the paper had already been shown in the no-shift setting, so understanding whether there is a simple technique that enables adaptation of results from that setting to the covariate shift setting would yield some nice insight, and could better enable future work. However, I understand that the authors have space constraints to contend with.

Correctness: Yes.

Clarity: Yes, although the experiments section could be less terse.

Relation to Prior Work: Yes, relation to prior work is generally well-covered. It might be nice to have references to the non-covariate-shift versions of reported results for the theorems if they exist.

Reproducibility: Yes

Additional Feedback: As mentioned above, some insight into the central technique for the proofs would be nice to have in the main text. I would also like some insight into the comparison between the oracle behavior policy case and the other cases where the behavior policy is unknown. Why doesn’t the known behavior policy estimator achieve the efficiency bound? Is this result already known in the no-shift case? It would be nice to see how this translates to the finite-sample settings. Could the authors add this as an estimator in the experiments? The DM and IPW methods are listed as requiring Donsker conditions. Would this still be the case with cross-fit versions of these estimators? My understanding is that cross-fitting can be used to remove this condition generally, but I may be wrong about this. If so, it seems unfair to claim that the singly-robust methods require such a condition, but the DR methods do not. In reality, these are sort of orthogonal factors. It might make sense, for example, to compare the DR estimator without cross fitting to the DR estimator with cross fitting in the experiments. On the subject of Donsker conditions, it would be nice to have a sentence describing how previous methods required this condition, and a quick statement of broadly what this condition entails. It sort of comes out of nowhere in the exposition. I understand that I have asked for a number of elaborations that might not be possible to fit in the allotted space. One way to reclaim space might be to remove the argument demonstrating that the estimator is doubly robust; the “you only need to get one model right” assumption is, in many communities, viewed as sort of a red herring, with the rate properties of DR estimators being the more interesting property. If this were removed in favor of a table showing, e.g., the difference between cross-fit and non-cross-fit estimators using non-Donsker classes, I think this would be more enlightening. Some small notational nitpicks: - The notation denoting estimators from different cross-fit folds k is not consistent between the text and algorithm 1 (subscript vs parenthetical superscript). - The IPWCS and DM methods seem to have reversed meanings for the ‘-R’ suffix. Is one of the sentences reversed?


Review 2

Summary and Contributions: This paper studies off-policy evaluation under covariate shift. In particular, the authors give a nonparametric efficiency bound for estimating the mean reward of a policy in an evaluation population, given combined historical and (unsupervised) evaluation data. A doubly robust estimator is proposed which is shown to attain the bound asymptotically under usual n^(1/4) rate conditions. A corresponding algorithm for policy learning is also proposed, and simulation experiments show some finite sample results. ----- Update post-rebuttal: I agree with the other reviewers' comments about improving readability, and feel the authors have responded satisfactorily to my comments/suggestions.

Strengths: All claims appear sound, both theoretically and empirically. The work uses state of the art doubly robust estimation procedures, with solid grounding in semiparametric efficiency theory. I appreciate that the authors present the specific efficiency bound and gave some discussion and interpretation. Given the prevalence of covariate shift, and the relative lack of attention paid to statistical optimality elsewhere, I expect the work to be relevant to the NeurIPS community both in terms of application and methods.

Weaknesses: The main weakness I see is that the results are by now quite standard, with relatively little apparent theoretical advance. For example, the efficiency bound and proposed doubly robust estimator follow from essentially the same logic as in off-policy evaluation without covariate shift, albeit with an extra density ratio term. This is not to say the work is not practically important, of course.

Correctness: The claims and empirical methodology all appear correct.

Clarity: This paper is very well-written and clear, with a logical structure and flow.

Relation to Prior Work: I feel the authors appropriately cite previous work. As noted in Additional Feedback below, it is not so clear to me how substantially different this is from an iid mixture model setting.

Reproducibility: Yes

Additional Feedback: 1) It is not clear to me that the differences the authors stress in Remark 2 are that substantial: for example, in the iid case where one observes say S ~ Bern(rho), along with X when S=1 and Z when S=0, how would the efficiency bound change compared to that given in Theorem 1? 2) Since the authors are thinking about cases where rho is a fixed constant, potentially under investigators’ control, maybe it would make sense to say something about the optimal choice of rho, which I think will have a simple “ratio of standard deviations” interpretation. 3) It is not clear to me what the “informal” theorems in Section 5 actually contribute to the paper. These results omit numerous required conditions, which I think obscures the points and could confuse readers. I recommend removing these as theorems and instead just discussing the properties in a paragraph or two. For example, it seems to me that these results will require strong undersmoothing conditions on the kernel estimator, as well as smoothness conditions on the nuisance functions, neither of which are mentioned.


Review 3

Summary and Contributions: This paper considers a setting where a (non-sequential) policy must be evaluated/learned based on results from a historical policy. In contrast to prior work, the new policy is evaluated/learned on a covariate distribution that differs from the historical distribution. This covariate shift describes situations where a study has been done and its results are used in a different setting. To account for covariate shift, this paper adapts an existing estimator of the policy return by adding a term that weights each datapoint by the density ratio of the two covariate distributions. Furthermore, the authors show the asymptotic Cramer-Rao lower bound on the efficiency of policy return estimators in the present setting, and show that the proposed estimator achieves it. They then compare some theoretical properties of different estimators.

Strengths: The paper considers a realistic and important setting: a study has been performed in some population (e.g. university students) and its results are meant to be applied in a setting where covariates differ (e.g. age and education). An efficiency bound is shown for the present setting, which is a reasonable contribution. The proposed estimator usually performs well in a a bandit setting on two datasets.

Weaknesses: UPDATE: On 1): I believe the paper has a long way to go in terms of presentation and I hope the authors will take carefully restructure the paper as promised. On 2): The author feedback better explains the degree of novelty. On 3): The authors point out that my criticism only applies to OPL, not OPE, which I previously overlooked. Based on this, I updated my score to a 6. 1) Presentation: As many parts of the analysis are not properly explained, it is hard to evaluate this paper and I am left skeptical. For example, the explanation of theorem 7 is rather confusing. The comparison of estimators (section 5.2) just lists a few claims which are not supported. In most places, the paper is written as though what is being said is obvious. With a thorough re-write and focusing on the key points, this paper could be more convincing. 2) Novelty/relevance: The proposed estimator seems like a rather incremental extension of Shimodaira, 2000; Sugiyama et al., 2008. These papers also used importance weighting with the density ratio of covariate distributions. This is a well-known way to adapt estimators for covariate shift and, in my view, it is not necessary to write a new paper for each setting where this method could be used. Additionally, the above sources are not cited in the text where the estimator is introduced, which first led me to believe the contribution has a higher novelty than it does. If the goal is to bridge the gap to a scientific community where this method is not well-known, I would recommend finding a different publication venue than NeurIPS. If instead the main contribution is the efficiency bound, reviewers at a more statistical venue may be better placed to assess how important this contribution is. 3) Experiments: There should be a comparison to the optimal policy learned on the behavior data. In principle, this policy may do very well on the evaluation data too. Even if traditional estimators of the policy return are biased without adjusting for covariate shift, they may still lead to the correct optimal policy. If so, the value of an unbiased return estimator becomes unclear. In particular, if the model is well-specified, Sugiyama et al. 2009 would suggest that the optimal policy is the same with or without covariate shift.

Correctness: I did not find errors.

Clarity: As noted above, the way the paper is written detracts from its usefulness. For example, the introduction reads more like a short version of the paper instead of helping the reader understand its relevance and key points.

Relation to Prior Work: As noted above, the relation to other papers about estimators under covariate shift is not clearly shown. Furthermore, the related work section doesn’t make it very clear what is the most closely related work and how this paper builds on it. E.g. “our perspective is not the same as theirs” should be more specific. Additionally, the section doesn’t seem to address any work on OPL/OPE.

Reproducibility: Yes

Additional Feedback: “Under a covariate shift, standard methods of OPE do not yield a consistent estimator of the expected reward over the evaluation data.Moreover, a covariate shift changes the efficiency bound of OPE, which is the lower bound of the asymptotic mean squared error (MSE) among reasonable n-consistent estimators.”: references would be helpful. The introduction doesn’t explain very clearly why covariate shift is a problem for OPE/OPL (other than with the claim above, but the reasons should be clarified). “Note that we can assume that samples are i.i.d in our problem by treating ρ is a random variable and assuming each replication follows a mixture distribution.” - please explain this more clearly Please ensure that Table 2 with caption is self-contained. "the OPE methods under the covariate shift have not been researched well, and people apply standard OPE methods to cases under the covariate shift." - This is an important claim (given in the broader impact section) which should be referenced and could then be used to support the importance of the paper.


Review 4

Summary and Contributions: This paper considers the off-policy evaluation problem when, in addition to a shift in the conditional distribution of action given policy, there is a shift in the observed covariates themselves. The proposed solutions involves an additional density ratio which is an importance sampling weight from the observed to target domain. The authors also examine the behavior of the proposed estimator within a double robust estimator. After reading the authors' response I have upgraded my score. I still feel that the writing should be improved, but am satisfied with the rest of the authors' responses and have. updated accoringly.

Strengths: * I think the problem setting is very interesting and important. * The use of density ratio (while not novel to this work) is a nice solution to the problem. * The analysis provide a set of interesting results.

Weaknesses: * I found the contributions of the paper to be difficult to follow. It seems the authors’ claimed contribution is (1) the use of an additional weight to adjust for covariate shift, (2) the use of direct density ratio estimation and, (3) asymptotic results showing double robustness. However, the paper as it stands makes it difficult to clearly read these. * The overlap conditions seem to be especially difficult to satisfy in this setting especially for high dimensional covariate sets where overlap violations become very likely (D’Amour, et al. 2017). * The contribution of adding an additional weight to account for covariate shift is an interesting idea but should be put in better context. The idea is similar to the transportability estimator put forth by Barenboim & Pearl (PNAS) & collaborators. Further, the use of direct density estimation for policy evaluation was also proposed recently (Sondhi, et al. (AISTATS 2020)). * It is not clear why two separate weights are required when using direct density ratio estimation. What prevents the ratio being learned for both weights simultaneously? * Typical off-policy evaluation with discrete actions use a rejection step in the estimator (c.f., Dudik, et al. (2011)). However, this does not appear to be mentioned anywhere in the paper or used in the experiments. In practice this can have a dramatic effect on the performance of estimators. It is difficult to parse and understand performance of the proposed method w.r.t. the other methods in light of this. * What is the motivation behind using Nadarya-Watson estimators for the methods compared to? We should expect this estimator to perform very poorly, even under an otherwise competitive alternative approach. Why not also allow those methods the use of direct density ratio estimation? * In the supplement results are included that use self-normalization (i.e., Hajek estimators). However, it appears that the unnormalized version of the proposed method greatly outperform the normalized version. This is incredibly counter intuitive to me, as normalization almost uniformly will improve estimators in practice on most datasets. What is the reasoning / intuition behind this. * The experiments all use 800 samples, this is (a) a _very_ small number and (b) does not give a sense of how any of the estimators will perform as sample size is increased. In practice, we would expect to see at least an order of magnitude more samples than this.

Correctness: The claims appear to be correct.

Clarity: I found this paper very difficult to read. Specifically: (1) The notation is cluttered and does not appear to follow standard notations. For example, the authors use `r` to denote the density ratio rather than the reward (which is common in the off-policy literature). (2) The layout of the paper makes it difficult to follow the authors' contributions. It would seem to me that there are two claimed contributions in this work (1) the use of a second density ratio estimate to account for distribution shift in (x) in addition to the shift in conditional distributions, and (2) a set of theoretical results which give the asymptotic analysis of the performance of a double robust estimator which takes advantage of (1). However, as the paper is currently laid out it is very difficult to parse those things apart.

Relation to Prior Work: As discussed above, the paper does make clear that the problem of distribution shift is considered (in contrast to prior work), however the authors should spend additional time clearly delineating and contextualizing the contributions of the work.

Reproducibility: Yes

Additional Feedback: I feel that this is the beginning of a very interesting paper. However, I would encourage the authors to spend additional time adding clarity to the writing and improving experimental results. **Update** Thank you to the authors for their response. After reading the response, and consulting with the other reviewers, i have upgraded my score. I encourage the authors to improve on the accessibility of the draft and the puzzling experimental results, but otherwise feel this is an interesting work.

[Author Response · NeurIPS 2020]

We appreciate the reviewer's constructive suggestions. Reviewer 1 and 2 highly evaluate our work. It looks the score of Reviewer 3 and 4 is mostly due to our paper's presentation style. We will sincerely take their suggestion into account in a revision phase. For each question, we answer as follows.

**Reviewer 1   Weakness:** We will try to add more insight into the theoretical approach, such as the difference between our case and non-covariate shift case. **Additional feedback:** (1) Using an oracle behavior policy is not enough to achieve the efficiency bound because we need a control variate. Estimating behavior policy in an unknown case is seen as using some control variate. We will add this explanation and improve the experiment; (2) Cross fitting versions of DM and IPW estimators do not have an efficiency property (Theorem 2) since these estimators do not have the doubly robust structure. We will also add the explanation of Donsker conditions more following your suggestion.

**Reviewer 2   Additional feedback**: (1) This is a good point. Unlike Theorem 1, we cannot obtain the explicit form of the efficiency bound under this i.i.d sampling; (2) Exactly. We will add this point; (3) Yes, we would follow your suggestions.

**Reviewer 3   Weakness**: (1) Due to the space constraint, we could not add a full explanation for each theoretical result. We will introduce a more intuitive explanation by rearranging the structure (we can make more space following Reviewer 1 and 2 suggestions); (2) Although an idea of IS itself is widely known, the analysis of IS based estimators is specific to each setting. We emphasize that this analysis is much more than the extension of this literature. Note that we cited covariate shift literature in the Introduction and Preliminaries. Here are the differences of Shimodaira (2000); Sugiyama et al. (2008) and ours. First, the goal is different. Our goals are OPE and OPL. On the other hand, their goal is solving regression problems or evaluating the expected log-likelihood. Second, due to the difference in goals, the analysis of the estimator is completely different. In OPE, we calculate the asymptotic first-order term of the estimator when plugging nonparametric estimators into the density ratio (Theorem 2). This implies that our analysis takes the plug-in effect into account. The general density ratio estimation literature such as Sugiyama et al. (2008); Shimodaira (2000) does not analyze this type of plug-in estimator though they actually use in practice. The effect of the plug-in is not negligible in OPE since the asymptotic variance is generally changed due to the plug-in. In this sense, our analysis is considered to be more sophisticated and tailored to an OPE problem. Third, for our OPE setting, we show not only the asymptotic distribution of estimators but also the efficiency bound. (3) For OPL, we agree with your opinion. We will incorporate it into the experimental section. On the other hand, in OPE, to estimate the policy value, we need to conduct covariate shift adaptation as our paper. **Relation to prior work**: This is due to space constraint since we try to refer to various literature as much as possible rather than explaining specific literature in detail. We will try to add more explanation. **Additional feedback**: We will add references in the parts you mentioned. The meaning of "Note that we can ..." is the i.i.d case where one observes $S \sim \mathrm{Bern}(\rho)$, along with $X$ when $S = 1$ and $Z$ when $S = 0$.

**Reviewer 4   Weakness**: (1) We appreciate the reviewer's detailed suggestion. We would try to make our contribution more clear following it; (2) We agree, but we also think this problem is a more general OPE problem rather than ours; (3) We will cite Sondhi, et al. (Note we already cited Barenboim & Pearl's work). (4) In Remark 2, we discussed a related technical difference between the shift of action and covariate. For the method, we do not need to estimate the density ratio simultaneously. For example, in the case where there are two actions $\{1, 2\}$, we estimate $p(a = 1 \mid x) = \frac{p(a=2)p(x|a=2)}{p(x)}$ and $\frac{q(x)}{p(x)}$. We can obtain $p(a = 2 \mid x)$ by $1 - p(a = 1 \mid x)$. If estimating the density ratio directly, we estimate $\frac{p(a=1)p(a=1|x)}{q(x)}$ and $\frac{p(a=2)p(x|a=2)}{q(x)}$. Thus, in both cases, **we need two estimators**. In addition, in general, **there is no significant difference between the convergence rates of** $p(a = 1 \mid x)\frac{q(x)}{p(x)}$ **and** $\frac{p(a=1)p(x|a=1)}{q(x)}$.. Moreover, before submission, we tried a method based on simultaneous estimation of the density ratio, but the performance was not good. We consider that the density ratio estimation is harder to fit nonparametrically than logistic regression. As a result, for ease of presentation, we did not adopt such a direction; (5) The purpose of rejection sampling step of Dudík et al. (2014) is to stabilize an estimator constructed from a dynamic and nonstationary policy, where $\pi^{\mathrm{e}}/\pi^{\mathrm{b}}$ can take an extreme value. In such a case, the technique will stabilize the estimator heuristically. However, there is a trade-off between bias and stabilization. Hence, in general, we do not have to use heuristics such as rejection sampling and clipping for all cases. (6) Firstly, we showed the experimental result using a direct density ratio estimator (IPWCS-R) as mentioned in L290. As you mentioned, it does not work in practice. Secondly, We introduced IPW-type estimators as a meta-estimator allowing any types of methods to estimate the density ratio. We analyze the specific IPW estimator when using the Nadarya-Watson estimator to estimate the density ratio since the asymptotic analysis is relatively straightforward. (7) We will investigate it. (8) As mentioned in L268, for OPE, we showed the experimental results with different sample sizes, 300, 500, 1000, in Appendix. We will add the results of OPE and OPL with more sample sizes in the next revision. **Clarity**: As far as our knowledge, $y$ is also used. I suppose the difference of this convention is due to the difference of communities, causal community, and RL community.

[Meta-Review · NeurIPS 2020]

The reviewers found the paper "interesting," "significant," "important," and "elegant," and I wholeheartedly agree with their assessment. For instance, Reviewer 1 says: "The thoroughness of results and clarity of exposition is impressive." Indeed, the masterfulness in applying semiparametric efficiency theory and the construction of doubly robust estimators are quite impressive. Even though some of the logic in these derivations is shared across the shifted and non-shifted (standard) domains, as noted by Reviewer 2, the application to this setting is quite compelling, and the results should be useful in practice. On the other hand, Reviewer 3 was a bit more critical regarding novelty but agrees that these results are applicable in "realistic and important" scenarios. Further, Reviewer 4 is positive about the paper while noting that the work should be put into context. In particular, the reviewer highlights that this is one specific instance of a transportability estimator (Bareinboim and Pearl, PNAS, 2016) (BP. henceforth). Even though there is a generic discussion of possible connections with the related literature, readers would benefit from a more detailed discussion and better articulation of what the paper is doing with respect to this broader context. In fact, this particular estimator holds when the source of variations regarding the covariates changes across settings while the mechanisms underlying the treatment and outcome variables remain invariant. In other words, the treatment A is S-ignorable w.r.t to outcome Y given the set of covariates A. The assumption of S-ignorability implies BP's Eq. 15, which seems to be equivalent to the re-weighting discussed in the paper (but for the relabeling of the variables). Transportability theory, on the other hand, delineates the conditions under which any type of extrapolation could be performed while guaranteeing consistency, and S-ignorability is just one among the viable identification strategies. The work in the paper is a significant first step. It leads to the interesting question of whether the new machinery developed could be extended to other transportability functionals to construct the corresponding double robust and efficient estimators. After all, consider reading the reviews carefully and consider their suggestions in the final version of the manuscript. All in all, this is an excellent piece of work, and my recommendation is "strong accept."